# GnT1IP-L specifically inhibits MGAT1 in the Golgi via its luminal domain

Hung-Hsiang Huang[1], Antti Hassinen[2], Subha Sundaram[1], Andrej-Nikolai Spiess[3], Sakari Kellokumpu[2], Pamela Stanley[1]*

[1]Department of Cell Biology, Albert Einstein College of Medicine, New York, United States; [2]Faculty of Biochemistry and Molecular Medicine, University of Oulu, Oulu, Finland; [3]Andrology Division, University Medical Center Hamburg-Eppendorf, Hamburg, Germany

**Abstract** Mouse GnT1IP-L, and membrane-bound GnT1IP-S (MGAT4D) expressed in cultured cells inhibit MGAT1, the N-acetylglucosaminyltransferase that initiates the synthesis of hybrid and complex N-glycans. However, it is not known where in the secretory pathway GnT1IP-L inhibits MGAT1, nor whether GnT1IP-L inhibits other N-glycan branching N-acetylglucosaminyltransferases of the medial Golgi. We show here that the luminal domain of GnT1IP-L contains its inhibitory activity. Retention of GnT1IP-L in the endoplasmic reticulum (ER) via the N-terminal region of human invariant chain p33, with or without C-terminal KDEL, markedly reduced inhibitory activity. Dynamic fluorescent resonance energy transfer (FRET) and bimolecular fluorescence complementation (BiFC) assays revealed homomeric interactions for GnT1IP-L in the ER, and heteromeric interactions with MGAT1 in the Golgi. GnT1IP-L did not generate a FRET signal with MGAT2, MGAT3, MGAT4B or MGAT5 medial Golgi GlcNAc-tranferases. GnT1IP/*Mgat4d* transcripts are expressed predominantly in spermatocytes and spermatids in mouse, and are reduced in men with impaired spermatogenesis.

*For correspondence: pamela.
stanley@einstein.yu.edu

Competing interests: The authors declare that no competing interests exist.

## Introduction

The N-acetylglucosaminyltransferase MGAT1 (GlcNAc-TI or GnT-1) catalyzes the transfer of GlcNAc from UDP-GlcNAc to $Man_5GlcNAc_2Asn$ of glycoproteins in the medial Golgi to initiate the synthesis of complex and hybrid N-glycans (*Robertson et al., 1978*; *Tabas et al., 1978*; *Kornfeld and Kornfeld, 1985*). In experiments to identify the activity of murine cDNA 41334120*Rik*, two transcripts were characterized in the mouse (*Huang and Stanley, 2010*). The longer transcript encodes a membrane-bound protein that inhibits MGAT1 in transfected cells, and is termed G̲lcN̲Ac̲T̲-I̲ I̲nhibitory P̲rotein, L̲ong form, GnT1IP-L (*Huang and Stanley, 2010*). GnT1IP-L is a Type II membrane glycoprotein with sequence homology to glycosyltransferase genes in family 54 in the CaZY database (*Cantarel et al., 2009*). A rat testis membrane-bound form has been termed GL54D but its activity has not been determined (*Au et al., 2015*). The mouse homologue of GL54D is the shorter transcript, previously termed GnT1IP-S (*Huang and Stanley, 2010*), and recently designated MGAT4D by the Human Genome Nomenclature Committee. When the N-terminus of GnT1IP-S is extended by a Myc or HA tag, it becomes membrane-bound and inhibits MGAT1 in cultured cells, similar to GnT1IP-L. In male germ cells mouse GnT1IP-S is probably membrane-bound like its rat homologue GL54D (*Au et al., 2015*). The sequence of GnT1IP-L (Genbank accession HM067443) is identical to GnT1IP-S with an additional 44 N-terminal amino acids.

Our previous study (*Huang and Stanley, 2010*) showed that transfection of a cDNA encoding GnT1IP-L inhibits endogenous or co-transfected MGAT1 activity in Chinese hamster ovary (CHO) cells. However, cell lysates of GnT1IP-L transfectants with low MGAT1 activity exhibit normal levels of B4GALT1 and MGAT3 activities (the latter in LEC10 CHO cells [*Campbell and Stanley, 1984*]). Co-immunoprecipitation

**eLife digest** Proteins are made up of chains of amino acids that fold into three-dimensional shapes and many are assembled in a cell compartment known as the endoplasmic reticulum. From here, these new proteins move to another compartment called the Golgi, where they may be further modified before they are transported to their final destination in the cell.

One way that proteins may be modified is known as glycosylation, in which sugar molecules are attached to specific amino acids. Some sugar molecules can act as labels that ensure the new proteins are transported to the correct destination in the cell. For proteins that are delivered to the surface of the cell, the sugar molecules can also play important roles in communication with other cells.

A simple sugar molecule, or a complex arrangement of many sugar molecules, may be attached to an amino acid by glycosylation. An enzyme called MGAT1 controls the synthesis of sugars called complex N-glycans in the Golgi. In 2010, researchers reported that a glycoprotein called GnT1IP-L binds to MGAT1 and inhibits its activity, thereby blocking the production of complex N-glycans. GnT1IP-L was found in the endoplasmic reticulum and Golgi, but it was not clear how it inhibits MGAT1.

Huang et al.—including some of the researchers from the 2010 study—have now investigated the activity of GnT1IP-L in cells grown in the laboratory using several biochemical techniques. The experiments show that GnT1IP-L only binds to MGAT1 when both proteins are in the Golgi. There are three sections (or 'domains') in GnT1IP-L, but Huang et al. found that only the domain that is on the inside of the Golgi is involved in this interaction.

Previous work indicated that GnT1IP-L may be involved in the formation of sperm in mice. Huang et al. have now analyzed previously published data on samples of testis tissue from human patients and found that the gene that encodes GnT1IP-L is present in very low amounts in patients whose sperm do not develop properly.

Huang et al.'s findings suggest that GnT1IP-L may inhibit MGAT1 to control the glycosylation of proteins in the Golgi of developing sperm. The next step is to test this hypothesis by generating mutant mice that lack GnT1IP-L, or to make GnT1P-L in other cells in which it is not normally made, to find out if this affects the production of sperm.

experiments showed that GnT1IP-L interacts physically with MGAT1, but does not interact with trans Golgi B4GALT1, nor the trans Golgi network sialyltransferase, ST8SIA2. Deletion mutagenesis experiments showed that removal of 39 amino acids from the C-terminus of membrane-bound Myc-GnT1IP-S, or removal of the stem domain from Myc-GnT1IP-L, abrogates MGAT1 inhibitory activity (*Huang and Stanley, 2010*).

In this paper, we investigate whether GnT1IP-L inhibits MGAT1 via its luminal or cytoplasmic and transmembrane (TM) domain, and whether GnT1IP-L retained in the endoplasmic reticulum (ER) can inhibit MGAT1. The specificity of GnT1IP-L for MGAT1 compared to related medial Golgi GlcNAc-transferases, and the interactions of GnT1IP-L in the ER and Golgi, were investigated by dynamic fluorescent resonance energy transfer (FRET) and bimolecular fluorescence complementation (BiFC) experiments that previously identified homomeric and heteromeric interactions between Golgi glycosyltransferases (*Rivinoja et al., 2009*; *Hassinen et al., 2010*, *2011*; *Hassinen and Kellokumpu, 2014*). The combined results show that GnT1IP-L inhibitory activity lies in its luminal domain, that it forms homomers in the ER, and in the Golgi it forms heteromers specifically with MGAT1. Interestingly, data extracted from published RNA-seq and microarray experiments reveal differential and complementary expression of mouse *Mgat1* and GnT1IP/*Mgat4d* genes in male Sertoli and germ cells, and show that transcripts of human GnT1IP/*MGAT4D* are markedly reduced in testis biopsies of men with impaired spermatogenesis.

## Results

### GnT1IP-L inhibits MGAT1 via its luminal domain

To investigate whether the TM or luminal domain of GnT1IP-L is important for inhibition of MGAT1 in CHO cells, different mutant and chimeric expression plasmids were constructed (*Figure 1* and *Table 1*). Constructs were transfected into CHO cells and stable populations selected for hygromycin

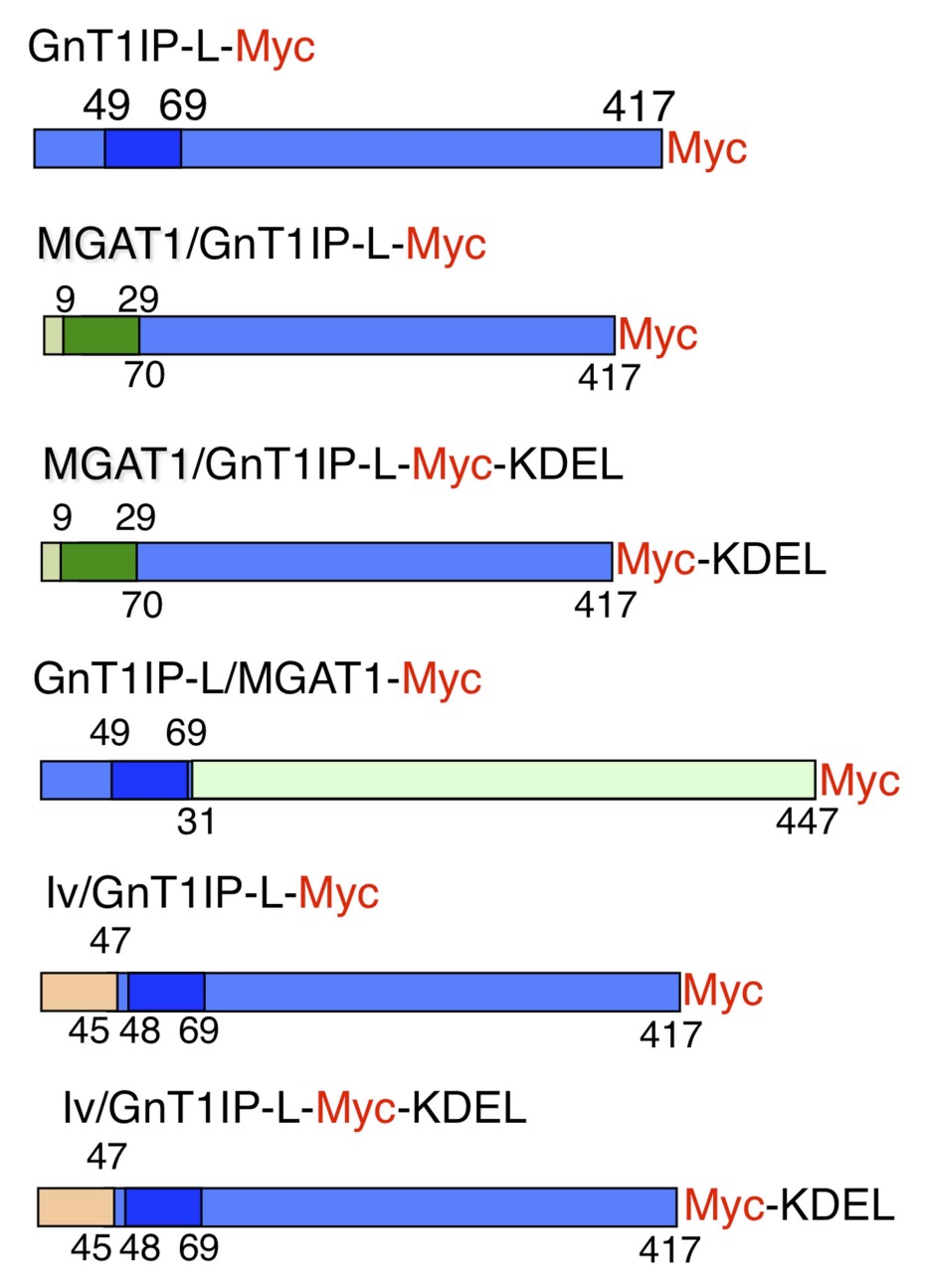

**Figure 1**. Expression constructs. Mouse GnT1IP-L (417 aa) contains an N-terminal cytoplasmic domain of 48 aa, a transmembrane (TM) domain of 21 aa (shaded), and a luminal domain of 348 amino acids. The location of the Myc tag (red) is shown for each construct. Chimeric constructs contained the cytoplasmic and TM domain of MGAT1 (green) linked to the luminal domain of GnT1IP-L (blue), or the cytoplasmic and TM domain of GnT1IP-L linked to the luminal domain of MGAT1, or N-terminal aa 1–47 of human Invariant chain p33 (Iv; beige) linked to aa 45 to 417 of GnT1IP-L. Predicted TM domains are shown in darker colors. Numbers on top of each chimera are aa from the N-terminal domain and underneath are aa from the luminal domain.

resistance were examined for resistance to the toxicity of *Phaseolus vulgaris* leukoagglutinin (L-PHA), and/or binding of the lectin *Galanthus nivalis* agglutinin (GNA). Resistance to L-PHA, accompanied by increased expression of cell surface oligomannose N-glycans detected by GNA, are hallmarks of inhibition of MGAT1 activity in CHO cells (*Chen and Stanley, 2003*; *Huang and Stanley, 2010*). The subcellular localization of each construct was investigated by transient transfection of HeLa cells and

**Table 1**. Primers for expression constructs

GnT1IP-L-Myc

| |
|---|
| For: 1301: (*HindIII,* Kozak) CAGATC*AAGCTT*CCACCATGTGCCTGGGAGAAAGTGTTGGGGACC |
| Rev: 1346: (**Myc**, *BamH1*) GACTAG*GGATCCC*TA**CAGATCCTCTTCTGAGATGAGTTTTTGTTC**GTAATAATTATCCTTGAGGTGC |

Myc-GnT1IP-L

| |
|---|
| For: 1312: (*HindIII,* Kozak, **Myc**) CAGATC*AAGCTT*CCACCATG**GAACAAAAACTCATCTCAGAAGAGGATCTG**TGCCTGGGAGAAAGTGTTGGGGACC |
| Rev: 1313: (*BamH1*) GCCTGT*GGATCCC*TAGTAATAATTATCCTTGAGGTGCTG |

HA-GnT1IP-L

| |
|---|
| For: 1068: (*HindIII,* Kozak, **HA**-GnT1IP-L) GGAACT*AAGCTT*CCACCATG**TACCCTTATGACGTCCCCGATTACGCCAGCCTG**TGCCTGGGAGAAAGTGTTGGGG |
| Rev: 1313 |

GnT1IP-L (F/<u>L</u>)

| |
|---|
| For: 1431: CTC**TTA**GCC**TTA**GTTGCCGTCCTGCTC**TTA**GGT**TTA**TCGTGT**TTA**TGC |
| Rev: 1432: GCATAAACACGATAAACCTAAGAGCAGGACGGCAACTAAGGCTAAGAG |

GnT1IP-L (F/<u>A</u>)

| |
|---|
| For: 1467: CTC**GCC**GCC**GCC**GTTGCCGTCCTGCTC**GCT**GGT**GCC**TCGTGT**GCC**TGC |
| Rev: 1468: GCAGGCACACGAGGCACCAGCGAGCAGGACGGCAACGGCGGCGGCGAG |

GnT1IP-L-Myc-KDEL

| |
|---|
| For: 1301 |
| Rev: 1471: (*BamH1* Myc-KDEL) *GGATCCC*TACAACTCATCTTT**CAGATCCTCTTCTGAGATGAG** |

MGAT1/GnT1IP-L-Myc

| |
|---|
| For: 1282 (*HindIII,* Kozak) GGACCG*AAGCTT*CCACCATGCTGAAGAAGCAGTCTGCAGGGC |

Internal: MGAT1/GnT1IP-L

| |
|---|
| Rev: 1434: TTGATTATTGGTTTGGTTCATCCTCCAGAAGAAGAGGAGCAGCAG |
| For: 1433: CTGCTGCTCCTCTTCTTCTGGAGGATGAACCAAACCAATAATCAA |
| Rev: 1346: (*BamH1*, <u>Myc</u>) GACTAG*GGATCCC*TA**CAGATCCTCTTCTGAGATGAGTTTTTGTTC**GTAATAATTATCCTTGAGGTGC |

MGAT1/GnT1IP-L-Myc-KDEL

| |
|---|
| For: 1282 |
| Rev: 1471: (*BamH1*, <u>Myc-KDEL</u>) *GGAGTCGGATCCC*TA**CAACTCATCTTTCAGATCCTCTTCTGAGATGAG** |

Iv/GnT1IP-L-Myc

| |
|---|
| For: 1435: (*HindIII,* Kozak) GGACCG*AAGCTT*CCACCATGCACAGGAGGAGAAGCAGG |

Internal Iv/GnT1IP-L

| |
|---|
| Rev: 1436: CAAGTTAACGTTCTTGGCCTTCATTCCGCGGCTGCACTTGCTCTC |
| For: 1437: GAGAGCAAGTGCAGCCGCGGA**ATG**AAGGCCAAG**A**ACGTTAACTTG |
| Rev: 1346: (*BamH1*, Myc) |

Iv/GnT1IP-L-Myc-KDEL

| |
|---|
| For: 1435 |
| Rev: 1471 |

GnT1IP-L/MGAT1-Myc

| |
|---|
| For: 161: (Kozak) GCCACCATGTGCCTGGGAGAAAGTGTTGGGGACCTG |

Internal (MGAT1/GnT1IP-L)

| |
|---|
| Rev: 162: GTCTGAGGGCAGCCTGCCAGGTGCTGGGCGGGAGATGCAGAAACACGAGAAACCAAAGAG |
| For: 163: CTCTTTGGTTTCTCGTGTTTCTGCATCTCCCGCCCAGCACCTGGCAGGCTGCCCTCAGAC |
| Rev: 164: (MGAT1-**Myc**) CTA**CAGATCTTCTTCAGAAATAAGTTTTTGTTC**ATTCCAGCTAGGATCATAGCCAGTCCATGT |

analysis of immunofluorescence using antibodies to Myc or HA, Golgi α-mannosidase II (MAN2A1), or GM130, or ER protein disulfide isomerase (PDI). In initial experiments, five Phe residues in the GnT1IP-L TM domain were all replaced with either Leu (similar hydrophobicity index to Phe) or Ala (hydrophobicity reduced ∼50% compared to Phe or Leu). Transfectants expressing GnT1IP-L(F/L) or GnT1IP-L(F/A) (Table 1) at similar levels based on western analysis, had an increased ability to bind GNA, and exhibited resistance to the toxicity of L-PHA (Figure 2B and data not shown). Thus, replacement of five Phe residues with Ala in the TM domain of GnT1IP-L did not markedly reduce its MGAT1 inhibitory activity.

To investigate the GnT1IP-L luminal domain, the TM and cytoplasmic domains of GnT1IP-L were replaced with the cytoplasmic and TM domains of MGAT1 to create the construct MGAT1/GnT1IP-L-Myc (Figure 1 and Table 1). The chimeric protein was localized to the Golgi compartment (Figure 2A), was well expressed, and conferred resistance to L-PHA in stable CHO transfectant populations (Figure 2B,C). The L-PHA resistance assay in Figure 2B shows transfectants or control cells that were stained by methylene blue after ∼3 days of growth from 2000 cells plated in the presence of increasing concentrations of L-PHA. Plates were stained when wells incubated in medium alone (no L-PHA) had become confluent. The variability seen in the proportion of transfectants highly resistant to L-PHA in populations expressing GnT1IP-L mutant or chimeric proteins is due to variable expression levels of cDNAs and is also observed with wild-type GnT1IP-L (see Figure 5B; Huang and Stanley, 2010). The important parameter is the proportion of cells in a transfectant population that consistently resist the toxicity of L-PHA. Homogenous mutant Lec1 CHO cells that completely lack MGAT1, or cells selected for high expression of GnT1IP-L (Huang and Stanley, 2010), are uniformly resistant to L-PHA (Figure 2B).

When a C-terminal KDEL retention sequence (Cancino et al., 2013) was added to the MGAT1/GnT1IP-L-Myc chimera, resistance to L-PHA was reduced (Figure 2B), consistent with reduced localization to the Golgi (Figure 2A). This result suggests that the luminal domain of GnT1IP-L is responsible for its ability to inhibit MGAT1. An important control was to examine the reverse chimera—the cytoplasmic and TM domains of GnT1IP-L linked to the luminal domain of MGAT1, termed GnT1IP-L/MGAT1-Myc (Figure 1 and Table 1). This chimera did not cause stable transfectants to become resistant to L-PHA (Figure 3A), and did not induce hypersensitivity to Con A (Figure 3B), in two independent clones with equivalent expression (Figure 3C). In addition, the activity of MGAT1 in the GnT1IP-L/MGAT1-Myc transfectant lysates was 6.1 or 15.5 nmol/mg protein/hr, respectively, compared to 7.7 nmol/mg/hr in a CHO cell lysate and 0.5 nmol/mg protein/hr in a Lec1 lysate. The activity of B4GALT1 in the same lysates was equivalent (16–21 nmol/mg protein/hr). A separate experiment with the same extracts gave qualitatively similar results. The fact that one GnT1IP-L/MGAT1-Myc transfectant did not have increased MGAT1 activity may reflect the efficiency of active enzyme formation when the chimeric protein was overexpressed. Nevertheless, it is clear that GnT1IP-L/MGAT1-Myc does not significantly inhibit MGAT1 activity whereas MGAT1/GnT1IP-L is inhibitory. Thus, the GnT1IP-L luminal domain is active when localized by the MGAT1 cytoplasmic and TM domain, and the luminal domain of GnT1IP-L is necessary to inhibit MGAT1 activity.

## GnT1IP-L is a specific inhibitor of MGAT1

MGAT1 is a resident of the medial Golgi, along with other GlcNAc-transferases of the N-glycan pathway MGAT2, MGAT3, MGAT4 and MGAT5. Previous experiments have shown that GnT1IP-L does not inhibit MGAT3 activity, although it interacted with MGAT3 in immunoprecipitation assays in CHO cell lysate. MGAT1 and MGAT2 form a complex (Nilsson et al., 1993, 1994; Opat et al., 2000), and their interactions have been directly observed in a dynamic FRET assay (Hassinen et al., 2010, 2011; Hassinen and Kellokumpu, 2014). To determine if GnT1IP-L inhibits medial Golgi glycosyltransferases other than MGAT1, CHO stable transfectants expressing HA-GnT1IP-L were assayed for MGAT2 and MGAT5 activities. A specific acceptor for MGAT4 was not available for assay. Relative expression was determined as a ratio to B4GALT1 activity, which is not affected by GnT1IP-L (Huang and Stanley, 2010). It can be seen in Table 2 (Source code 1) that whereas GnT1IP-L inhibited MGAT1 activity by ∼75%, MGAT2 and MGAT5 activities were not markedly inhibited in CHO cells. We also found that GnT1IP-L inhibits MGAT1 activity in COS-7 cells, and induces increased expression of GNA binding reflecting increased expression of oligomannose N-glycans at the surface of COS-7 cells, as expected when MGAT1 is inhibited (data not shown).

To further investigate the specificity of GnT1IP-L for MGAT1, a dynamic FRET assay was employed. Previous assays of glycosyltransferases tagged at the C-terminus by monomeric cerulean (mCer) or monomeric venus (mVen) determined FRET interactions by flow cytometry and showed that numerous glycosyltransferases of the N- and O-glycan pathways form homomers in the ER and heteromers in the

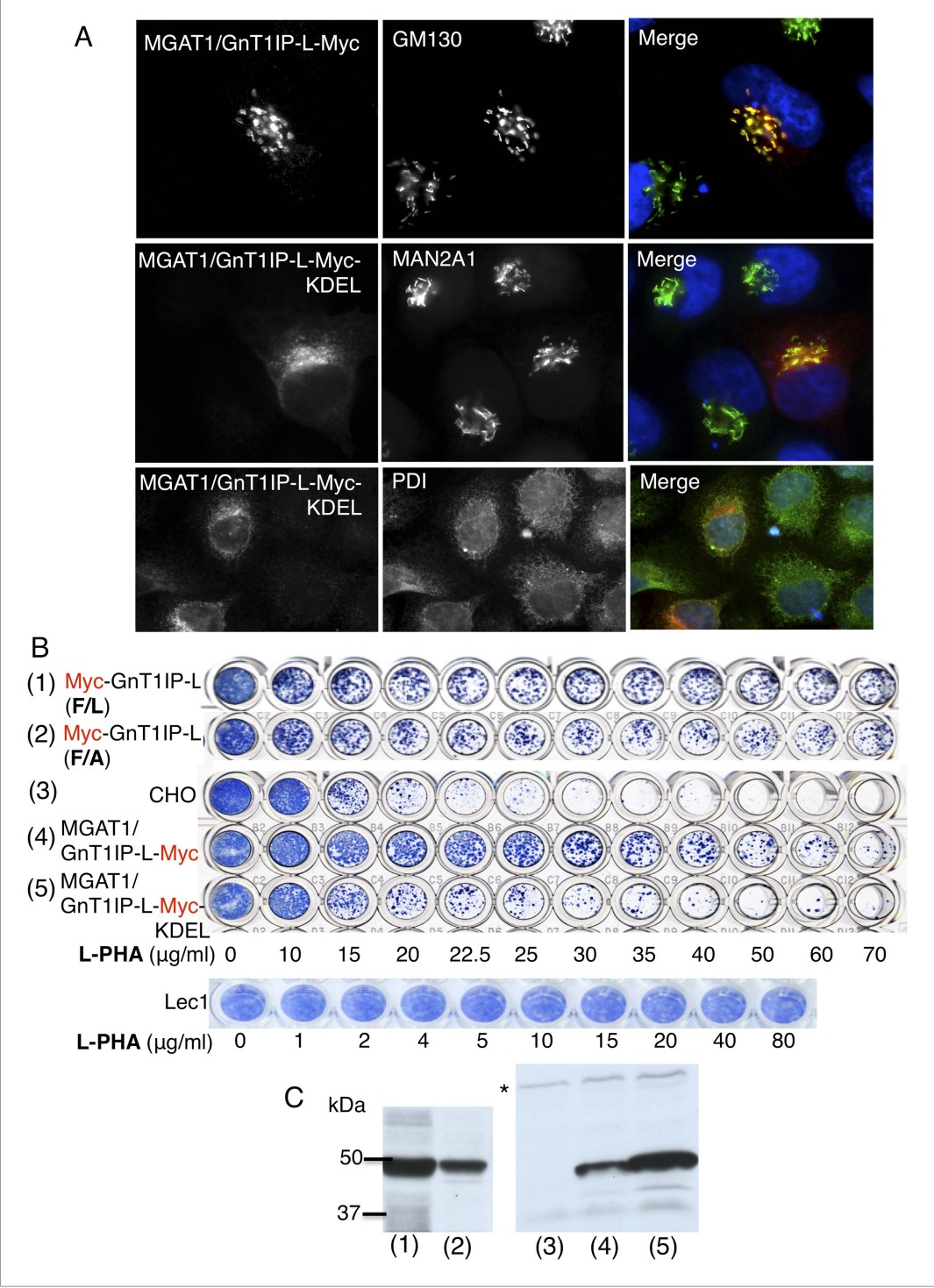

**Figure 2**. The luminal domain of GnT1IP-L inhibits MGAT1. (**A**) HeLa cells transiently expressing the chimera MGAT1/GnT1IP-L-Myc or MGAT1/GnT1IP-L-Myc-KDEL were analysed for expression of Myc, MAN2A1, GM130 and protein disulfide isomerase (PDI). Each result is representative of 40–50 cells examined. (**B**) Resistance to L-PHA of the same chimeric proteins along with Myc-GnT1IP-L(F/L) and Myc-GnT1IP-L(F/A) in Chinese hamster ovary (CHO) transfectant populations selected for hygromycin resistance, compared to CHO and Lec1 cells. Independent transfectant populations gave the same results in 2–3 replicate assays. (**C**) Western analysis of lysates corresponding to CHO populations numbered in panel **B**. The blot was probed with anti-Myc antibodies. * non-specific band loading control.

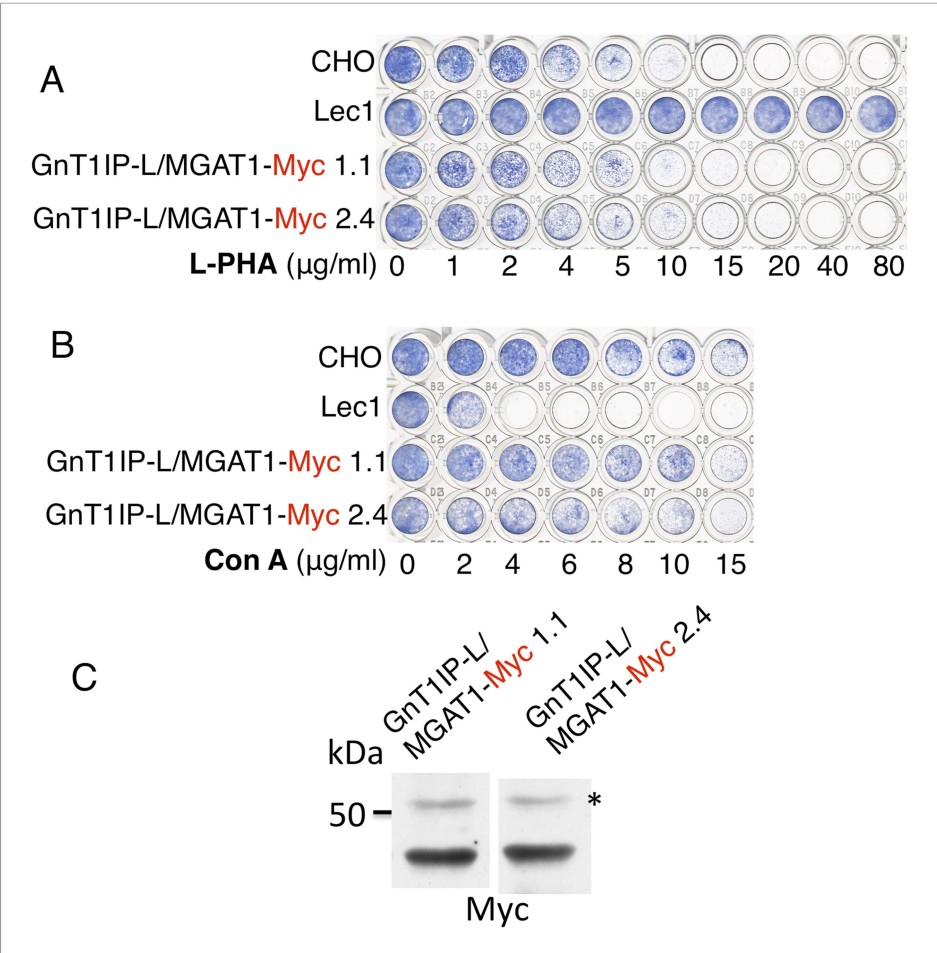

**Figure 3**. The TM and cytoplasmic domain of GnT1IP-L does not inhibit MGAT1. (**A**) Lectin-resistance test of cloned CHO cells stably expressing GnT1IP-L/MGAT1-Myc compared to CHO cells and Lec1 CHO cells that lack MGAT1 (n = 2). (**B**) The same cloned GnT1IP-L transfectant lines were compared to CHO and Lec1 cells for resistance to Con A (n = 2). (**C**) Western analysis of CHO cell lysates from the cloned transfectants in (**A**) and (**B**). * non-specific band shows equal loading.

Golgi (*Rivinoja et al., 2009*; *Hassinen et al., 2010*, *2011*; *Rivinoja et al., 2012*; *Hassinen and Kellokumpu, 2014*). The latter heteromeric interactions occur between glycosyltransferases that act sequentially in the same glycan pathway and in the same compartment of the Golgi. Interactions between GnT1IP-L and medial Golgi GlcNAc-transferases MGAT1 to MGAT5 were investigated by the same methods via transient transfection into COS-7 or Lec1 CHO cells stably expressing GnT1IP-L, or transient co-transfection of GnT1IP-L cDNA with an individual MGAT cDNA. Mouse GnT1IP-L-mVen was investigated for interactions with mouse or human MGAT1, MGAT2, MGAT3, MGAT4B and MGAT5. First, it was shown by fluorescence microscopy that each of the human GlcNAc-transferases tagged with mChe localized correctly to the Golgi when transfected into COS-7 or Lec1 CHO cells stably expressing GnT1IP-L-mVen (*Figure 4A* and data not shown). Measurements of FRET efficiencies revealed that GnT1IP-L interacted with MGAT1 but not with MGAT2, MGAT3, MGAT4B or MGAT5 in COS-7 or Lec1 CHO cells which lack endogenous MGAT1 (*Chen and Stanley, 2003*) (*Figure 4B* and *Figure 4—source data 1*). The same results were obtained with the mouse GlcNAc-transferases (*Figure 4C* and *Figure 4—source data 1*). The lower FRET efficiencies of mouse enzymes may reflect species differences as lower expression of all mouse MGAT constructs was observed. If the data are expressed as a percentage of the MGAT1/GnT1IP-L interaction, no differences are evident between mouse and human interactions. These data also show that GnT1IP-L forms homomers with itself as well as heteromers with MGAT1. The specificity of the FRET signal between GnT1IP-L and

**Table 2.** Glycosyltransferase activities in CHO cells expressing HA-GnT1IP-L

| Cells | B4GALT1 nmol/mg/hr | MGAT1 nmol/mg/hr | MGAT2 nmol/mg/hr | MGAT5 nmol/mg/hr |
|---|---|---|---|---|
| CHO Ratio to B4GALT1 | 15.6 (11–23.8) – | 4.2 (3.2–5.5) 0.27 | 0.41 (0.4–0.42) 0.026 | 0.34 (0.35–0.48) 0.022 |
| CHO/HA-GnT1IP-L Ratio to B4GALT1 | 11.5 (9.7–13.2) – | 1.1 (0.9–1.3) 0.096 | 0.38 (0.16–0.61) 0.033 | 0.33 (0.24–0.41) 0.028 |
| Ratio activity GnT1IP-L:CHO | **0.74** | **0.26** | **0.93** | **0.97** |

Glycosyltransferase assays were performed as described in 'Materials and methods' on cell extracts from CHO cells and CHO cells stably expressing HA-GnT1IP-L. Each assay was performed in duplicate, and activity (with range) is given as the average of duplicates for 2–3 independent assays.

MGAT1 was further demonstrated by inhibition of complex formation by overexpression of either GnT1IP-L-HA or MGAT1-HA (*Figure 4D* and *Figure 4—source data 1*). As expected, overexpression of HA-tagged MGAT2, MGAT3, MGAT4B or MGAT5 did not inhibit the FRET signal induced by co-transfection of GnT1IP-L and MGAT1 (*Figure 4D*).

MGAT1 and MGAT2 have previously been shown by FRET analyses to form heteromers in the Golgi (*Hassinen et al., 2010*, *2011*; *Hassinen and Kellokumpu, 2014*). To determine if GnT1IP-L inhibits formation of MGAT1/MGAT2 heteromers, tagged human or mouse MGAT1 and MGAT2 were transiently expressed with competitive GnT1IP-L-HA in COS-7 cells, and FRET efficiencies measured. All proteins localized to the Golgi (*Figure 4E*), and the presence of GnT1IP-L-HA did not interfere with the formation or stability of MGAT1/MGAT2 heteromers (*Figure 4F* and *Figure 4F—source data 1*).

## ER-localized GnT1IP-L does not inhibit nor interact with MGAT1

When GnT1IP-L is overexpressed, Golgi-localized MGAT1 is markedly relocated to the ER (*Huang and Stanley, 2010*). To determine if GnT1IP-L interacts with MGAT1 in the ER prior to exit for the Golgi, chimeric proteins were constructed using the N-terminal ER retention signal from human invariant chain p33 (termed Iv) (*Nilsson et al., 1991*), with and without a C-terminal KDEL retention sequence. Transfection of Iv/GnT1IP-L-Myc into HeLa cells gave predominant expression in the ER, whereas co-transfected MGAT1-HA was largely localized to the Golgi compartment (*Figure 5A*). Expression in wild type CHO cells was robust but did not lead to resistance to L-PHA when either Iv/GnT1IP-L-Myc or Iv/GnT1IP-L-Myc-KDEL were overexpressed in CHO cells (*Figure 5B,C*). Therefore, GnT1IP-L that is largely localized to the ER does not inhibit MGAT1 activity.

MGAT1, MGAT2 and other glycosyltransferases undergo homomeric interactions in the ER and heteromeric interactions in the Golgi (*Hassinen et al., 2011*; *Hassinen and Kellokumpu, 2014*). This paradigm was also investigated for GnT1IP-L and MGAT1 using BiFC analysis with N-terminal Venus (VN) or C-terminal Venus (VC) fragments attached to the C-terminus of MGAT1 or GnT1IP-L. Expression and Golgi localization of VN and VC fusion proteins were confirmed by staining with rabbit anti-GFP Ab (detected with anti-rabbit Ab conjugated to Alexa Fluor 594), and goat anti-GFP Ab (detected with anti-goat Ab conjugated to Alexa Fluor 405). Confocal microscopy was performed with a filter set that detected BiFC signal (green), VN signal (red) and VC signal (blue). To examine complex formation in the ER, COS-7 cells were treated with the microtubule disruptor nocodazole for 8 hr to prevent exit from the ER, as previously described (*Hassinen and Kellokumpu, 2014*). The effect of nocodazole treatment on interactions between GnT1IP-L-VN and GnT1IP-L-VC is seen in *Figure 6A*. Compared to the control in which both proteins were localized in the Golgi, nocodazole treatment (added 8 hr post-tranfection for 16 hr), caused both to accumulate in the ER. A BiFC signal was observed indicating GnT1IP-L homomer formation (*Figure 6A*, Nocodazole). In the absence of nocodazole (Control), homomers were transported to the Golgi. By contrast, when GnT1IP-L–VN was co-transfected with MGAT1-VC, a BiFC signal was not detected in nocodazole-treated cells (*Figure 6B*), indicating that heteromer formation did not take place in the ER (*Figure 6B*). However, in the absence of nocodazole (*Figure 6B*, Control), GnT1IP-L-VN and MGAT1-VC were transported to the Golgi with the concomitant emergence of the BiFC signal due to heteromer formation.

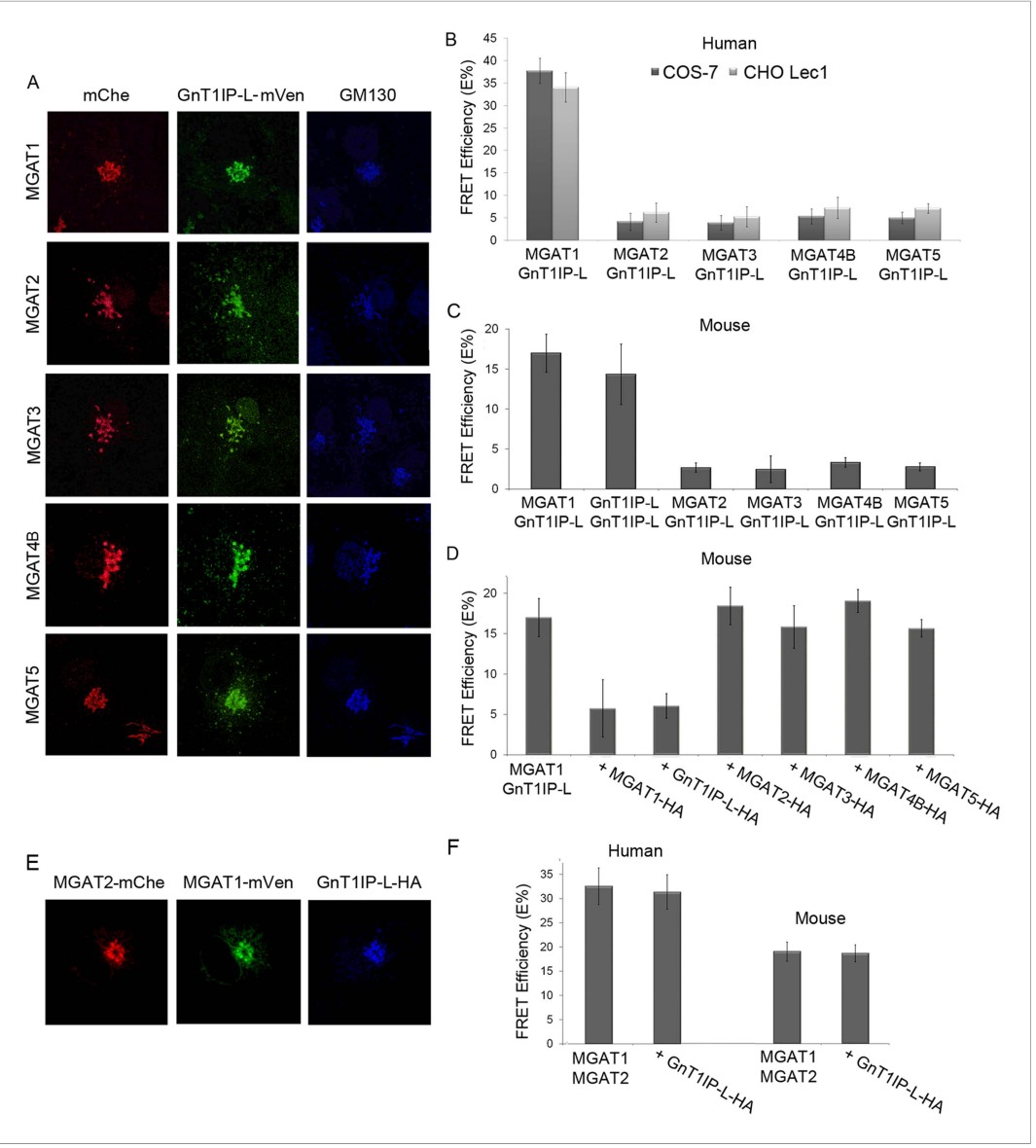

**Figure 4**. GnT1IP-L interacts specifically with MGAT1 in the Golgi. (**A**) Fluorescence microscopy of COS-7 cells stably expressing GnT1IP-L-mVen and transiently expressing medial Golgi GlcNAc-transferases MGAT1 to MGAT5 conjugated to mChe at their C-terminus compared to the Golgi marker GM130. (**B**) GnT1IP-L interaction with human GlcNAc-transferases. COS-7 or Lec1 CHO cells stably expressing GnT1IP-L-mVen were transfected with human cDNAs encoding MGAT1 to MGAT5 C-terminally tagged with mChe, and fluorescent resonance energy transfer (FRET) efficiencies were determined. (**C**) GnT1IP-L interaction with mouse GlcNAc-transferases. COS-7 cells transiently expressing mouse GnT1IP-L-mVen and GnT1IP-L-mChe or mouse MGAT1-mChe, MGAT2-mChe, MGAT3-mChe, MGAT4B-mChe or MGAT5-mChe and FRET efficiencies determined. (**D**) COS-7 cells stably expressing GnT1IP-L-mVen were co-transfected with mouse MGAT1-mChe together with competitive cDNA encoding mouse MGAT1 to MGAT5 C-terminally tagged with HA. (**E**) Transiently co-expressed MGAT1-mVen, MGAT2-mChe and GnT1IP-L-HA are localized in the Golgi. (**F**) COS-7 cells were transiently expressed with MGAT1-mVen, MGAT2-mChe and GnT1IP-L-HA, and FRET efficiencies were determined. Bars represent the mean ± STDEV (n = 10 cells).

The following source data is available for figure 4:

**Source data 1**. GnT1IP-L interactions with human and mouse MGATs in the Golgi of COS-7 and CHO Lec1 cells.

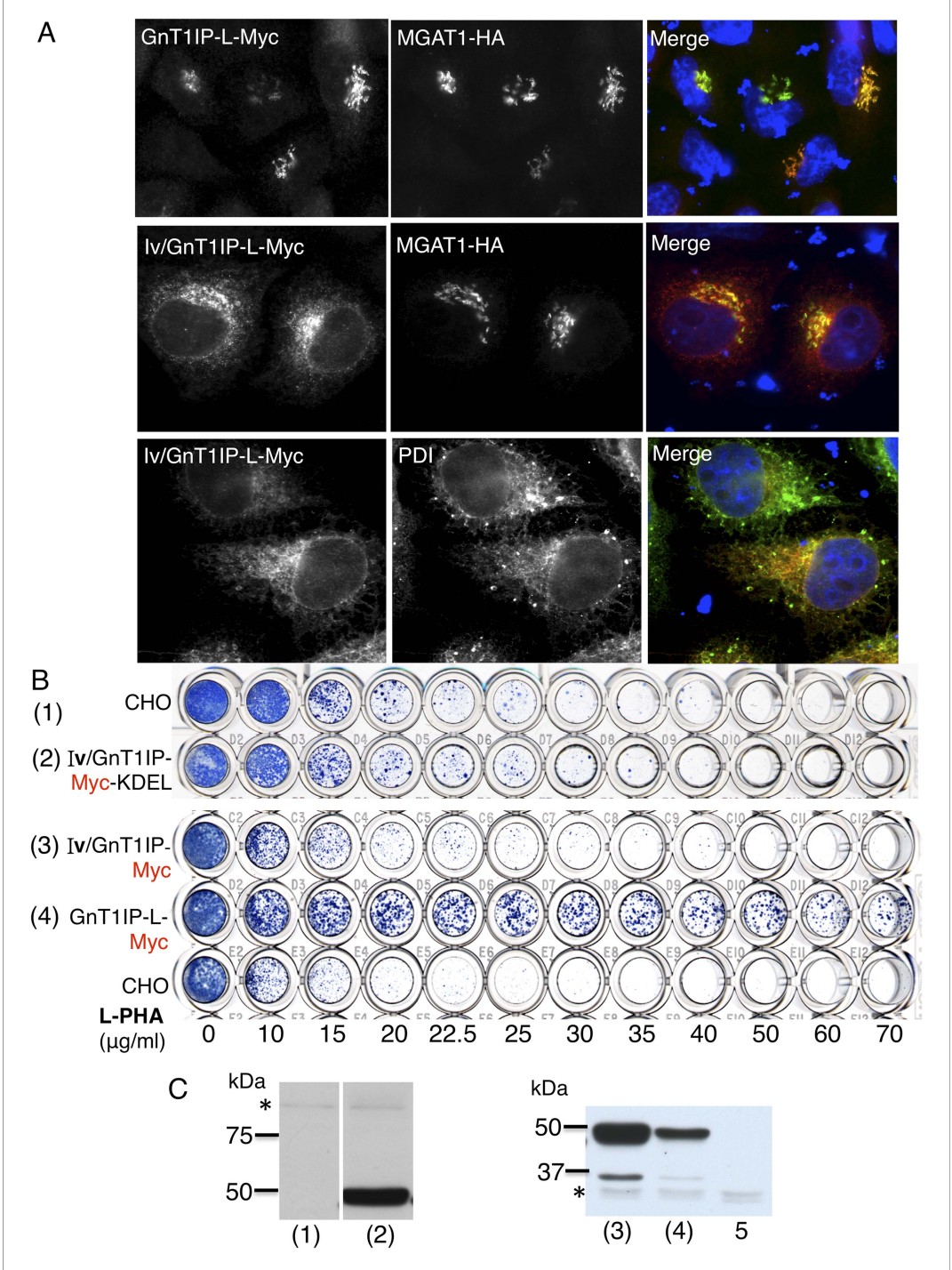

Figure 5. ER-localized GnT1IP-L does not inhibit MGAT1. (A) HeLa cells transiently expressing GnT1IP-L-Myc and MGAT1-HA, or the chimera Iv/GnT1IP-L-Myc with and without MGAT1-HA were analysed for expression of Myc, HA and PDI. Each result is representative of 40–50 cells examined. (B) Lectin-resistance test comparing the various GnT1IP-L stable transfectant populations with CHO cells for resistance to L-PHA. (C) Western analyses of CHO and stable transfectant populations expressing GnT1IP-L chimeric proteins probed with anti-Myc antibody. Lanes are numbered according to the corresponding cell populations in panel B. Lanes (1) and (2) were cropped from the same blot. Lane 5 is CHO cells expressing MGAT1-HA. * non-specific band loading control.

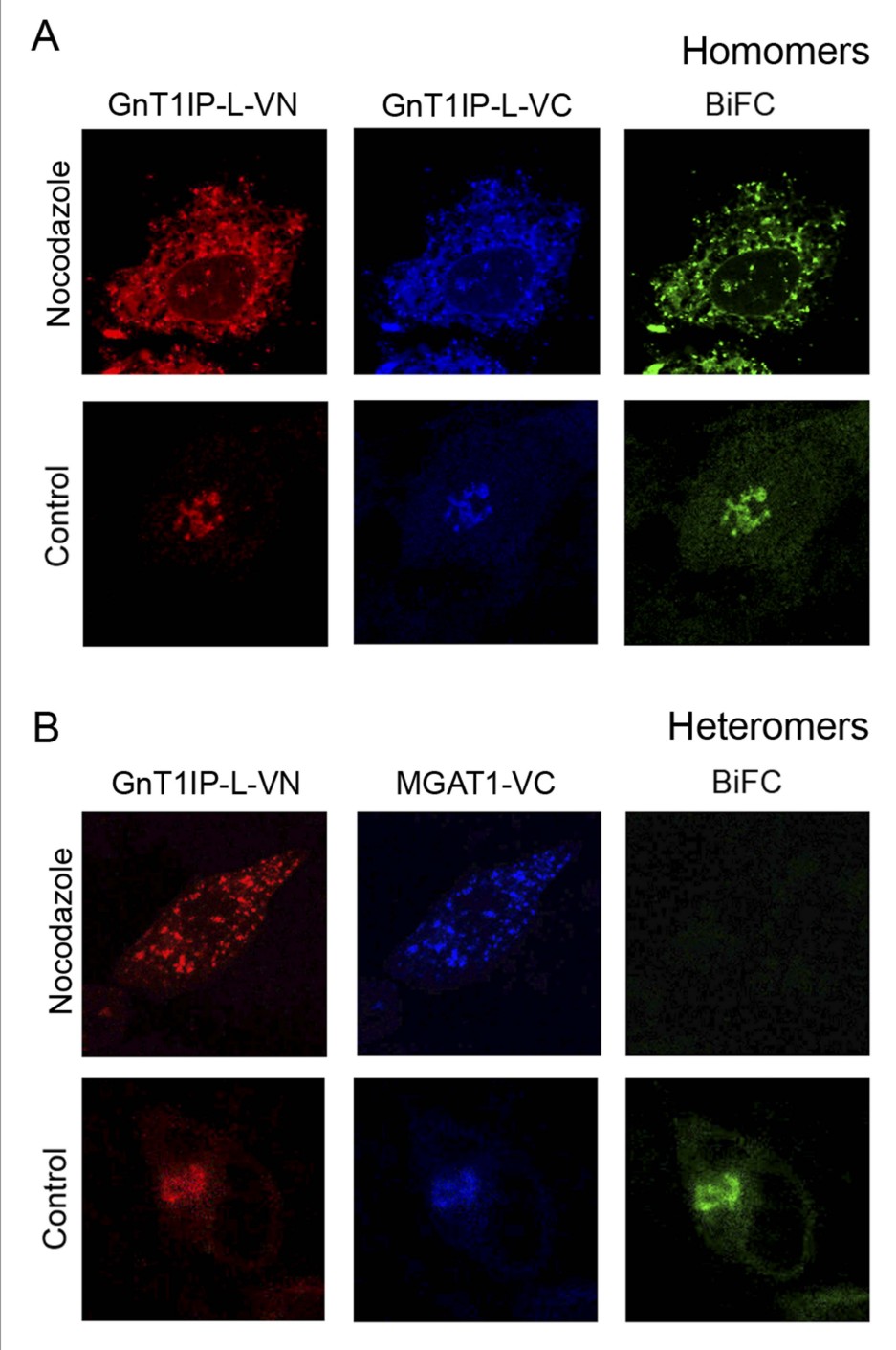

**Figure 6**. GnT1IP-L and MGAT1 form homomers in the endoplasmic reticulum (ER) and heteromers in the Golgi. (**A**) COS-7 cells were co-transfected with mouse GnT1IP-L-VN and GnT1IP-L-VC and, after 8 hr in culture, one set of plates was treated with 1 µg/ml nocodazole overnight. Cells were examined by fluorescence microscopy (50 cells/view) for expression and bimolecular fluorescence complementation (BiFC) signal. The VN tag was detected with rabbit anti-GFP and anti-rabbit Ab conjugated to Alexa Fluor 594, and the VC tag was detected with goat anti-GFP and anti-goat Ab conjugated to Alexa Fluor 405. Confocal imaging detected the BiFC signal (green), VN signal (red) and VC signal (blue). (**B**) COS-7 cells were co-transfected with mouse GnT1IP-L-VN and human MGAT1-VC, treated with nocodazole as in (**A**), and examined by fluorescence microscopy for expression and BiFC signal. All cells expressing GnT1IP-L-VN with GnT1IP-L-VC gave a BiFC signal in the presence and absence of nocodazole. In contrast, no signal was detected in nocodazole-treated cells expressing GnT1IP-L-VN with MGAT1-VC.

To further investigate GnT1IP-L/MGAT1 heteromer assembly, we utilized the dynamic FRET assay. COS-7 cells co-transfected with GnT1IP-L-mVen and MGAT1-mChe were treated with nocodazole at 16 hr post-transfection. Golgi-localized heteromers of GnT1IP-L and MGAT1 (*Figure 7A*, Control) were relocated to the ER (*Figure 7A*, Nocodazole) with a concomitant reduction of the FRET signal (*Figure 7B* and *Figure 7—source data 1*). Removal of nocodazole allowed reformation of heteromeric complexes in the Golgi (*Figure 7A,B*, Recovery). The histogram shows that, compared to the GnT1IP-L/MGAT1 heteromeric FRET signal, nocodazole treatment reduced heteromer formation, and removal of nocodazole partially rescued heteromer formation, whereas GnT1IP-L homomers were not affected by nocodazole treatment (*Figure 7B*).

Previous experiments showed that increasing the pH of the Golgi by ∼0.4 units following incubation for 4–16 hr in chloroquine inhibits heteromer formation and favors homomer formation between glycosyltransferases (*Hassinen and Kellokumpu, 2014*). To determine if Golgi acidity is also important for the formation of GnT1IP-L heteromers with MGAT1, COS-7 cells stably expressing GnT1IP-L-mVen were co-transfected either with GnT1IP-L-mChe or MGAT1-mChe, and treated with 40 µM chloroquine for 4 hr (added 16 hr post-transfection), or for 16 hr (added at 8 hr post-transfection). Compared to untreated controls, either treatment with chloroquine did not significantly reduce GnT1IP-L homomers, but caused an ∼60% reduction in GnT1IP-L/MGAT1 heteromers (*Figure 8A* and *Figure 8—source data 1*). To evaluate whether this reduction in heteromers is accompanied by an increase in the amount of GnT1IP-L homomers, MGAT1-HA and GnT1IP-L-mChe were co-transfected into cells stably expressing GnT1IP-L-mVen, and treated with chloroquine. The proportion of homomers increased almost twofold, presumably due to the disruption of heteromer formation (16 hr treatment), or their disassembly (4 hr treatment) at the higher Golgi pH (*Figure 8B* and *Figure 8—source data 1*). Chloroquine treatment did not impair the Golgi localization of any of the test proteins (*Figure 8C*). Therefore GnT1IP-L, like MGAT1 as shown previously (*Hassinen and Kellokumpu, 2014*), forms homomers in the ER and heteromers with MGAT1 in the acidic Golgi lumen, where it inhibits MGAT1 activity.

## GnT1IP transcripts are poorly expressed in testis biopsies from men with impaired spermatogenesis

We previously identified a potential function for GnT1IP-L in testis based on the observation that cells expressing GnT1IP-L, Myc-GnT1IP-S or lacking MGAT1 bind more tightly to a Sertoli cell line (*Huang and Stanley, 2010*). Lectin histology experiments in mouse and rat have shown that spermatocytes bind low levels of L-PHA (*Jones et al., 1992*; *Batista et al., 2012*), reflecting low expression of complex N-glycans due potentially to inhibition of MGAT1 by GnT1IP-L (*Jones et al., 1992*; *Batista et al., 2012*). The GnT1IP/*Mgat4d* gene is very highly expressed in mouse testes compared to all other tissues (see *Mgat4d* BioGPS microarray data [*Wu et al., 2009*, *2013*]). In mouse germ cells, expression of GnT1IP/*Mgat4d* based on microarray and RT-PCR data is very low in spermatogonia, highest in spermatocytes and intermediate in spermatids (*Chalmel et al., 2007*; *Huang and Stanley, 2010*). This expression pattern in mouse germ cells is complementary to *Mgat1* that is high in spermatogonia, and greatly reduced in spermatocytes (*Chalmel et al., 2007*). Very similar results are evident from an analysis of mouse RNA-Seq data that we interrogated for GnT1IP/*Mgat4d* and *Mgat1* transcripts (Gene Expression Omnibus Dataset GSE43717; [*Soumillon et al., 2013a*, *2013b*]). Mapping the relative expression values of GnT1IP/*Mgat4D*, (ENSMUSG00000035057) and *Mgat1* (ENSMUSG00000020346) onto the expression values of all 36,823 transcripts for different mouse germ cell subtypes clearly indicates that GnT1IP/*Mgat4D* (*Figure 9*, blue) is exclusively expressed in post-meiotic germ cells (*Figure 9—source data 1*). In contrast, *Mgat1* (*Figure 9*, red) is expressed at lower levels in all germ cell types, as well as somatic Sertoli cells. These results, as well as the observation that antibodies to rat GnT1IP (GL54D) detect signals in spermatocytes and spermatids but not spermatogonia (*Au et al., 2015*), suggest post-meiotic transcriptional activation of the GnT1IP/*Mgat4d* gene. Interestingly, examination of the Soumillon et al. RNA-Seq data for the 130 nucleotides upstream of the *Mgat4d* start site which encode the sequence specific to GnT1IP-L, revealed very low numbers of reads that were not significant (data not shown). This may reflect the regulated expression of GnT1IP-L during spermatogenesis (*Iguchi et al., 2006*; *Huang and Stanley, 2010*).

GnT1IP/*MGAT4D* is also very highly expressed in human testis compared to 26 other tissues examined by RNA-Seq (ArrayExpress E-MTAB-1733 MGAT4D ENSG00000205301 [*Fagerberg et al., 2013*, *2014*]). In another study, *MGAT4D* transcripts were shown to be highly enriched in human testis

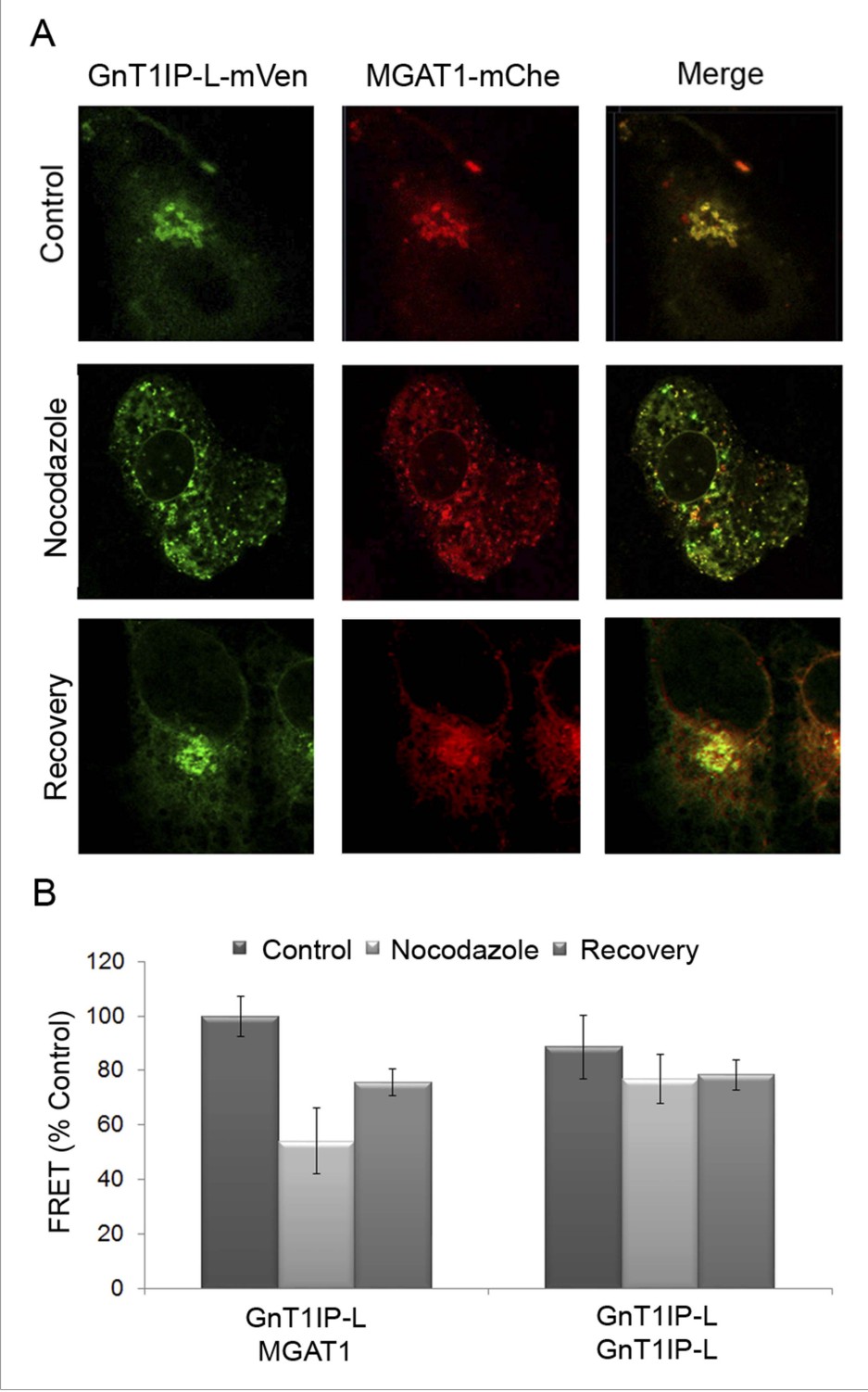

**Figure 7**. Golgi-localized GnT1IP-L and MGAT1 heteromers are disrupted in the ER following nocodazole treatment and reform after recovery. (**A**) COS-7 cells stably expressing GnT1IP-L-mVen were transfected with human MGAT1-mChe and, after 16 hr, treated with 1 µg/ml nocodazole. After 4 hr of treatment, nocodazole was removed from half the samples and recovery allowed to occur for 4 hr. Samples were examined by fluorescence microscopy. (**B**) FRET efficiencies of COS-7 cells stably expressing GnT1IP-L-mVen and transfected with either MGAT1-mChe or GnT1IP-L-mChe (as in **A**) were determined by FRET microscopy. FRET efficiencies (mean ± STDEV; n = 10 cells) are given as % of control. Control samples (100%) gave a FRET efficiency of 38 ± 3%.

*Figure 7. Continued*

The following source data is available for figure 7:

**Source data 1**. Disruption of Golgi-localized GnT1IP-L and MGAT1 heteromers in the ER following nocodazole treatment and their recovery in the Golgi after drug removal.

(29 fragments per kilobase of transcript per million mapped reads (FPKM) compared to a maximum FPKM of 0.1 for *MGAT4D* transcripts in 26 human tissues [*Djureinovic et al., 2014*]). To determine GnT1IP/*MGAT4D* expression in human germ cell subtypes, we investigated microarray data (ArrayExpress E-TABM-234) (*Cappallo-Obermann et al., 2008*) of human testis biopsies from men with different testicular phenotypes of impaired spermatogenesis (*Spiess et al., 2007*) with respect to the expression of GnT1IP/*MGAT4D* and *MGAT1*. The data show that GnT1IP/*MGAT4D* transcripts are very poorly expressed in all testicular phenotypes in which there are no pre-meiotic germ cells in the germinal epithelium (Tubular atrophy, TA; Sertoli cell only syndrome, SCO), or only pre-meiotic germ cells (only spermatogonia present (SPG)) (*Figure 10A*; *Figure 10—source data 1*). It is also evident that the phenotype of meiotic arrest (MA), which in the majority of cases occurs at the level of pachytene spermatocytes in the human, exhibits no significant GnT1IP/*MGAT4D* expression. However, testicular phenotypes presenting with reduced (hypospermatogenesis, HYS) or normal and unimpaired levels of round and elongated spermatids (full spermatogenesis, FS), display a massive increase in GnT1IP/*MGAT4D* expression. These findings point to a clear post-meiotic expression of GnT1IP during human spermatogenesis that occurs earliest at the level of secondary spermatocytes (mitotic phase of meiosis) or spermatids.

By contrast, *MGAT1* transcripts concomitantly decrease with increasing germ cell differentiation (*Figure 10B*), with highest expression in testicular phenotypes without germ cells (TA, SCO). This indicates an expression largely restricted to testicular somatic cell types. A small increase at the level of spermatogonia suggests that, in humans, *MGAT1* is expressed in spermatogonia whereas GnT1IP/*MGAT4D* is not, tallying with the data obtained from mouse microarray studies (*Chalmel et al., 2007*). The overall decline of *MGAT1* expression throughout spermatogenesis reflects a typical somatic transcript dilution effect (compare Figure 1 in *Cappallo-Obermann et al., 2013*), due to increasing numbers of germ cell-specific transcripts.

## Discussion

In this paper we show that the MGAT1 inhibitory activity of GnT1IP-L requires its luminal domain. Thus, mutations in the TM domain from Phe to Leu or Ala, or swapping the cytoplasmic and TM domain with that of MGAT1, do not significantly reduce GnT1IP-L inhibitor activity. The requirement for the luminal domain is consistent with our previous findings that removal of the C-terminal 39 aa of membrane-bound GnT1IP-S, or the stem domain of GnT1IP-L, abrogate inhibitor activity (*Huang and Stanley, 2010*). The specificity of GnT1IP-L for MGAT1 vs other GlcNAc-transferases of the medial Golgi was investigated here using BiFC and a dynamic FRET assay. These experiments showed no significant FRET activity between GnT1IP-L and MGAT2, MGAT3, MGAT4B or MGAT5. The only substantial FRET signal was obtained between GnT1IP-L and MGAT1, and this signal could only be inhibited by overexpression of either GnT1IP-L or MGAT1. As this result implies, and as shown previously for MGAT1, GnT1IP-L forms homomers with itself, as well as heteromers with MGAT1. These interactions were further defined using BiFC and FRET experiments following treatment with nocodazole or chloroquine. The combined data show that GnT1IP-L preferentially forms homomers in cells treated with nocodazole when it is confined to the ER, and heteromers with MGAT1 following nocodazole removal and a recovery period when it moves to the Golgi. GnT1IP-L homomers are also formed preferentially when the Golgi pH is elevated by treatment with chloroquine. Therefore, GnT1IP-L behaves like the glycosyltransferases previously studied (*Hassinen and Kellokumpu, 2014*), interacting with itself in the ER and primarily with MGAT1 in the Golgi. It is interesting that GnT1IP-L showed no FRET interaction with MGAT2 which is predicted to be in a 'kin recognition' complex with MGAT1 (*Nilsson et al., 1993*, *1994*), and which has been shown to form heteromers with MGAT1 using the dynamic FRET assay used here (*Hassinen et al., 2010*, *2011*; *Hassinen and Kellokumpu, 2014*). GnT1IP-L also did not inhibit or disrupt the formation of MGAT1/MGAT2 heteromers in a competition

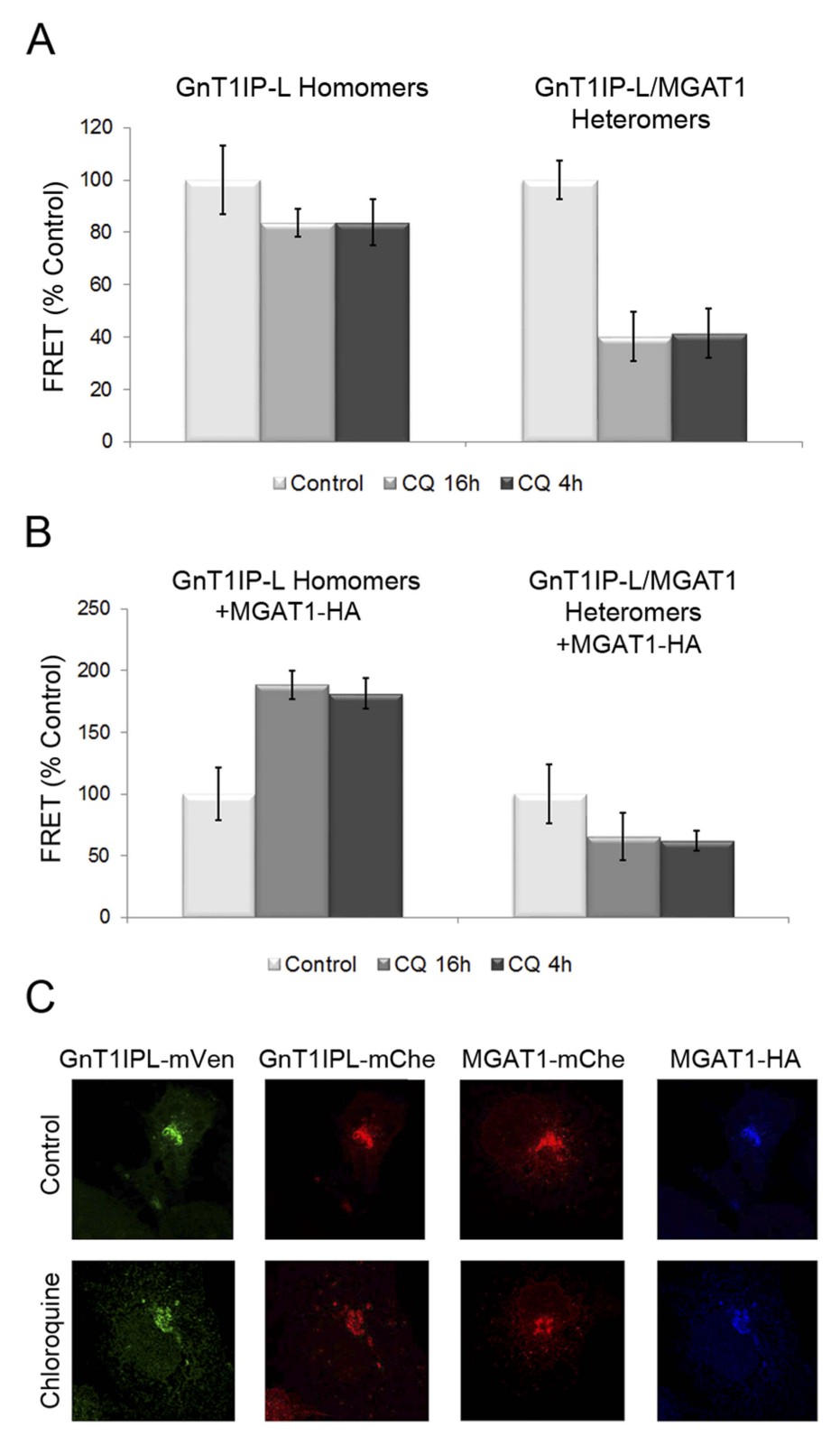

**Figure 8**. Chloroquine inhibits the formation of GnT1IP-L and MGAT1 heteromers and favors homomers. (**A**) COS-7 cells stably expressing GnT1IP-L-mVen were transfected with mouse GnT1IP-L-mChe or human MGAT1-mChe and treated after 8 or 16 hr with 40 µM chloroquine for 16 hr or 4 hr, respectively. Untreated control cells (100%) gave FRET efficiencies of 32% for the GnT1IP-L homomer and 38% for the heteromer with MGAT1. (**B**) The same

*Figure 8. continued on next page*

*Figure 8. Continued*

experiment as in (**A**) was performed except that, in addition, MGAT1-HA was added as a competitor. Bars in (**A**) and (**B**) show mean ± STDEV (n = 10 cells). (**C**) Fluorescence microscopy of the transfected cells in the presence or absence of 40 µM chloroquine.

The following source data is available for figure 8:

**Source data 1**. Disruption of GnT1IP-L and MGAT1 heteromers and enhanced formation of homomers following chloroquine treatment.

assay. Therefore, it may be concluded that when GnT1IP-L is in a complex with MGAT1, MGAT2 is not excluded from that complex, and that GnT1IP-L binds to a different site on MGAT1 than MGAT2. In addition, our data show that overexpression of MGAT2 did not disrupt GnT1IP-L/MGAT1 heteromers. The same lack of competition was observed for overexpression of MGAT3, MGAT4B and MGAT5.

A recent study of rat testis identified GL54D, the rat homologue of mouse GnT1IP-S, as the most abundant species amongst membrane proteins in Golgi preparations (*Au et al., 2015*). Immunohistochemistry showed that rat GL54D is confined to spermatocytes and spermatids, consistent with the expression pattern of GnT1IP transcripts in purified mouse germ cells (*Figure 9*; [*Soumillon et al., 2013a*]). Thus, while the GL54D homologue GnT1IP-S is secreted from CHO cells

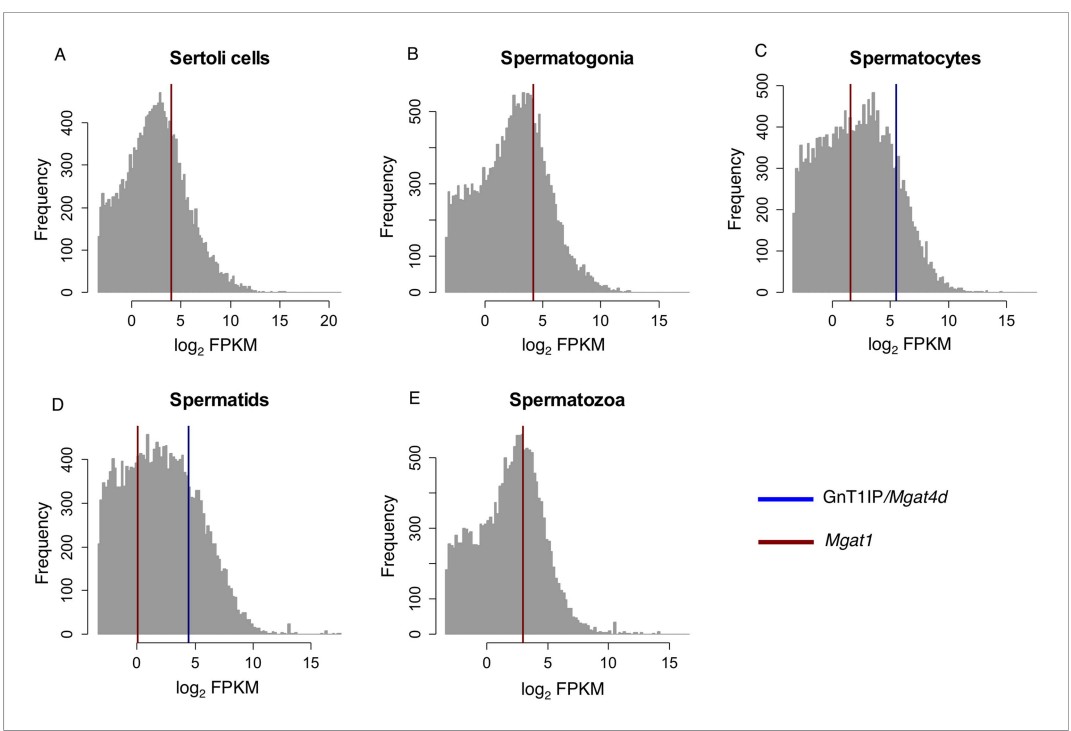

**Figure 9**. RNA-Seq data for GnT1IP/*Mgat4d* and *Mgat1* in mouse germ cells. Histogram overlay plot for GnT1IP/*Mgat4D* (blue) and *Mgat1* (red) gene expression in isolated mouse germ cell subtypes as described in *Soumillon et al. (2013a)*. (**A**) Sertoli cells, (**B**) Spermatogonia, (**C**) Spermatocytes, (**D**) Spermatids, (**E**) Spermatozoa. The grey histogram reflects the log$_2$-transformed **F**ragments **P**er **K**ilobase of transcript per **M**illion mapped reads (FPKM) values of all 36,823 transcripts identified by RNA sequencing and deposited in the GEO database as GSE43717. Red and blue overlayed vertical lines depict the expression values for *Mgat1* (ENSMUSG00000020346) and GnT1IP/*Mgat4D* (ENSMUSG00000035057), respectively. Note the absence of GnT1IP/*Mgat4D* transcripts in Sertoli cells, spermatogonia and spermatozoa.

The following source data is available for figure 9:

**Source data 1**. *R* code and comma-delimited data files for generating *Figure 9*.

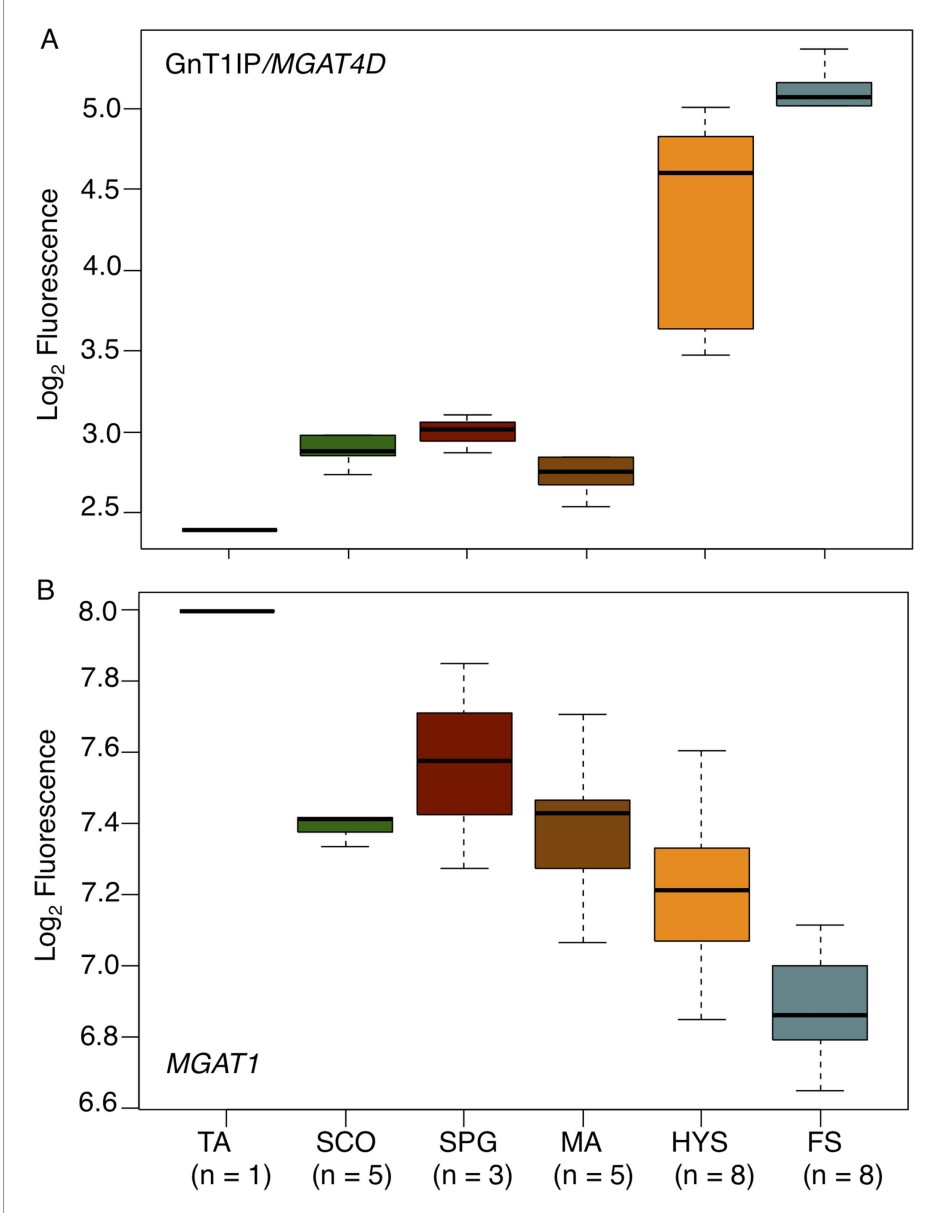

**Figure 10**. GnT1IP/*MGAT4D* and *MGAT1* transcripts in testis biopsies from men with impaired spermatogenesis. Transcript levels of GnT1IP/*MGAT4D* and *MGAT1* were determined from the microarray data of *Spiess et al. (2007)*. (**A**) Boxplot of GnT1IP/*MGAT4D* log₂ fluorescence in human testicular biopsies presenting with different types of spermatogenic impairment (tubular atrophy (TA), Sertoli cell only (SCO), presence of spermatogonial cells (SPG), meiotic arrest (MA) at the level of primary spermatocytes, hypospermatogenesis with decreased numbers of round/elongated spermatids (HYS), and full spermatogenesis with normal numbers of round/elongated spermatids (FS)). Sample size (n) for each group is given below the abscissa. (**B**) Same as in (**A**), but for *MGAT1*. Note the decreasing transcript abundance, which is a common observation for somatic transcripts in the presence of increasing germ cell content (*Cappallo-Obermann et al., 2013*).

The following source data is available for figure 10:

**Source data 1**. *R* code and comma-delimited data files for generating *Figure 10*.

(*Huang and Stanley, 2010*), it is likely to be membrane-bound in mouse germ cells, similar to GL54D in rat (*Au et al., 2015*).

While it is possible that GnT1IP may also have a glycosyltransferase activity, as suggested by its recent designation as MGAT4D, we have observed no evidence of such an activity in transfected cells.

In characterizing the CHO N-glycans generated following expression of membrane-bound GnT1IP, no complex N-glycans that might reflect GlcNAc transfer were identified by mass spectrometry, but rather a great increase in the abundance of the $Man_5GlcNAc_2$ substrate of MGAT1 was observed (*Huang and Stanley, 2010*). In addition, we have shown that GnT1IP-L induces increased binding of GNA reflecting enhanced expression of oligomannose N-glycans, increased resistance to lectins that bind complex N-glycans, and/or inhibition of MGAT1 activity, in a variety of cell lines including CHO, COS-7, HeLa cells (*Huang and Stanley, 2010*; this work) and PC3 cells (unpublished observations). Of course, a glycan, protein or lipid substrate may be present in only very low quantities or not expressed in CHO cells. Thus it cannot be ruled out that GnT1IP-L has an activity other than its ability to specifically inhibit MGAT1. Importantly however, another example of a gene that has homology to, and the protein domain structure of a glycosyltransferase, but an activity that is distinct, is C1GALT1C1, originally called COSMC (*Ju and Cummings, 2002*; *Wang et al., 2010*). This protein is a specific chaperone dedicated to the Gal-transferase C1GALT1, and essential for its activity (*Wang et al., 2010*). Thus, although GnT1IP-L (or MGAT4D) has homology to family 54 glycosyltransferases, the only activity yet identified for membrane-bound GnT1IP is as a specific inhibitor of MGAT1.

A functional role for GnT1IP-L has been proposed in testis based on the following: the gene encoding GnT1IP-L is most highly expressed in testis compared to other mouse tissues (*Wu et al., 2009*, *2013*; *Djureinovic et al., 2014*; *Fagerberg et al., 2014*), the expression of the transcript encoding GnT1IP-L is developmentally regulated (*Huang and Stanley, 2010*); GnT1IP is well expressed in spermatocytes and spermatids but not in spermatogonia (*Chalmel et al., 2007*; *Huang and Stanley, 2010*); and cells expressing GnT1IP-L and oligomannose N-glycans bind more tightly to TM4 Sertoli cells than cells expressing MGAT1 and complex N-glycans (*Huang and Stanley, 2010*). Most interestingly, the gene encoding MGAT1 is expressed in a complementary manner to GnT1IP-L in male germ cells (*Figure 9* and [*Chalmel et al., 2007*]). We are currently investigating the hypothesis that membrane-bound GnT1IP functions to down-regulate MGAT1 activity in spermatocytes and potentially spermatids, thereby enhancing their ability to bind to Sertoli cells. It is therefore of interest that men with impaired spermatogenesis exhibit greatly reduced expression of GnT1IP in microarray studies of testis biopsies (*Figure 10*). The degree of reduction appears to reflect the proportion of the remaining population of germ cells. Interestingly, MGAT1 transcripts were not reduced but were slightly increased reflecting robust expression of *MGAT1* in Sertoli cells (*Chalmel et al., 2007*). Germ cell expression of MGAT1 is however, essential for spermatogenesis in mice since conditional deletion of the *Mgat1* gene in spermatogonia blocks spermatogenesis and results in infertile males (*Batista et al., 2012*). Ongoing studies with conditional knockout mice, and mice overexpressing GnT1IP-L or MGAT1 in specific germ cell populations, should reveal roles for GnT1IP-L and MGAT1 in spermatogenesis.

## Materials and methods

### Plasmids

The plasmids Myc-GnT1IP-L, HA-GnT1IP-L, GnT1IP-L-Myc were prepared from mouse GnT1IP-L cDNA (accession number HM067443) using the primers given in *Table 1* that include *Hind*III or *BamH*1 restriction sites for insertion into pCDNA3.1 containing a hygromycin resistance cassette. The TM mutations F/L and F/A were made by site-directed mutagenesis using the primers shown in *Table 1* to generate Myc-GnT1IP-L (F/L or F/A) or HA-GnT1IP-L (F/L or F/A). Chimeric proteins were generated using a set of primers that included internal primers covering the boundaries of the sequences to be linked as shown in *Table 1*. The GnT1IP-L/MGAT1-Myc chimera was made similarly except that PCR fragments were subcloned into pStrata and the full length PCR product was cloned into pCDNA3.1 containing a zeomycin resistance cassette. All constructs were verified by DNA sequencing.

Full-length cDNA clones encoding mouse or human GlcNAc-transferases MGAT1 to MGAT5 were obtained from Imagenes GmbH (Berlin, Germany), or Open Biosystems Inc. (Huntsville, AL) (mouse *Mgat2* and *Mgat4b*) or cloned by us (mouse *Mgat1*; [*Kumar et al., 1992*]). Constructs for BiFC were pCDNA3-based and possessed C-terminal mYFP fragments VN or VC as described earlier (*Hassinen et al., 2010*, *2011*). FRET plasmids with C-terminal monomeric Venus (mVen) or monomeric mCherry (mChe), as well as cDNAs C-terminally tagged with HA or Myc, were prepared as described (*Hassinen et al., 2011*; *Hassinen and Kellokumpu, 2014*). All constructs were sequence-verified with the ABI3500xL Genetic Analyzer before use.

## Antibodies and lectins

Mouse anti-HA mAb (HA.11) and mouse anti-Myc mAb (9E10) were from Covance (Princeton, NJ), rabbit anti-HA polyclonal antibody (pAb) (Y-11) was from Santa Cruz Biotechnology Inc (Dallas, TX), mouse anti-beta actin mAb (AC-15) was from Abcam (Cambridge, MA), rabbit anti-human GM130 pAb was from EMD Millipore (Billerica, MA), mouse anti-rat Golgi GM130 mAb (35/GM130) was from BD Biosciences (San Jose, CA), goat horse radish peroxidase (HRP)-conjugated anti-mouse secondary antibody was from Thermo Fisher Scientific Inc. (Rockford, IL), rabbit anti-bovine PDI pAb and mouse anti-rat PDI mAb (1D3) were from Stressgen Biotechnologies Corp (San Diego, CA), and rabbit anti-human MAN2A1 pAb was a gift of Kelly Moremen (University of Georgia, GA). Secondary antibodies conjugated to Alexa-488 (green; goat anti-rabbit or anti-mouse), or Alexa-568 (red; goat anti-mouse IgG (H + L)) were from Invitrogen Life Technologies (Grand Island, NY). *P. vulgaris* leukoagglutinin (L-PHA), concanavalin A (Con A), GNA and GNA-FITC were from Vector Laboratories (Burlingame, CA).

## Cell culture, transfection and drug treatments

CHO cells were grown in suspension or on plates in alpha-modified Eagle's medium with 10% FBS (Gemini BioProducts, Sacramento, CA) in 5% $CO_2$ at 37°C. HeLa and COS-7 cells were grown on plates in Dulbecco's modified Eagle's medium with 10% FBS in 5% $CO_2$ at 37°C (HyClone, Thermo Scientific, Waltham, MA). Expression plasmids were transfected using FuGENE 6 (Promega Corp, Fitchburg, WI) according to the manufacturer's instructions. To obtain stable transfectant populations, antibiotic selection was initiated 24–48 hr post-transfection by adding ~$10^6$ transfectants to selection media containing 1 mg/ml active G418 (Gemini Bio-Products) for 5 to 7 days or 1.4 mg/ml hygromycin (EMD Millipore) for 1 day before switching to 0.7 mg/ml hygromycin for 4–6 days. Resistant colonies were pooled and characterized or sorted by fluorescence-activated cell sorting (FACS) for expression of GFP or binding of GNA-FITC prior to use. For FRET and BiFC experiments, cells cultured for 1 day were transfected using 0.5 µg of each plasmid cDNA and FuGENE 6 according to the supplier's protocol (Promega Corp). After 24 hr, cells were processed either for fluorescence microscopy, BiFC or FRET measurements (see below). Where noted, chloroquine (CQ) from Sigma Aldrich (St. Louis, MO) was added to the culture medium at 40 µM, or nocodazole (1 µg/ml, Sigma Aldrich) was added at different times as described in 'Results'.

## Fluorescence microscopy

For FRET and BiFC experiments, COS-7 and CHO cells were prepared for immunofluorescence microscopy as described previously (*Hassinen et al., 2010*). Briefly, after fixation with 4% paraformaldehyde for 15 min at room temperature, cells were permeabilized with 0.1% saponin in PBS and stained with anti-GM130 (BD Biosciences), mono- or polyclonal anti-HA (Sigma Aldrich), anti-FLAG (Sigma Aldrich), anti-Myc (Abcam), anti-PDI (5B5, M877, Dakopatts a/s, Denmark), rabbit anti-N-terminal GFP (Affinity Bioreagents, Golden, CO and goat anti-C-terminal GFP (Santa Cruz Biotechnology, Inc) antibodies. After washing, cells were treated with relevant Alexa fluor 405-, 488- and 594-conjugated anti-mouse, anti-rabbit and anti-goat secondary antibodies (Invitrogen, Carlsbad, CA. After staining, cells were mounted and imaged using the Zeiss LSM 700 confocal microscope, Zen2009 software (Carl Zeiss AG, Oberkochen, Germany), 63× or 100× Plan-Apo oil immersion objectives and appropriate filter sets for each dye.

For immunofluorescence experiments, HeLa cells ($3 \times 10^5$) were added to coverslips coated with poly-L-lysine in a 6-well dish and incubated at 37°C in 5% $CO_2$. After 16 hr cells were washed with PBS, fixed in 3% paraformaldehyde, and incubated in 0.2% Triton X-100, 1% FBS and 0.5% (wt/vol) bovine serum albumin (BSA, fraction V) in PBS containing 1 mM $CaCl_2$ and 1 mM $MgCl_2$ as described (*Huang and Stanley, 2010*). Following first and secondary antibody incubations in the same buffer, nuclei were stained with blue DAPI (1 µg/ml, Sigma–Aldrich). Coverslips were mounted using Fluoromount (SouthernBiotech, Birmingham, AL) and fluorescent images acquired on an inverted microscope (Zeiss Axiovert 200M) coupled to a 12-bit cooled charge-coupled device camera (Zeiss AxioCam MRm Rev. 3) controlled by Axiovision software (Zeiss AxioVs40, Version: 4.7.2.0), using a 100× 1.3 NA oil immersion objective (Zeiss EC Plan-NeoFluar), and saved as tif files (1388 × 1040, 8 bit).

## FRET and BiFC microscopy

FRET microscopy measurements were performed using the Zeiss LSM700 confocal microscope, mVen and mChe variants as the donor/acceptor FRET pair and the acceptor bleaching protocol with

appropriate filter sets for mVen and mChe (*Hassinen and Kellokumpu, 2014*). Samples were fixed before analysis as described for immunofluorescence (see above). The samples were subjected to iterative bleaching (30 cycles, 20 iterations, 555 nm, 70% laser intensity) during which the intensity values of the mVen were recorded. Background values were subtracted from the measured intensity values. FRET % was calculated from the acceptor-corrected intensities (a macro package from Zeiss) using the formula

$$\text{FRET } \% = \frac{(D_{post} - B_{post}) - (D_{pre} - B_{pre})}{(D_{post} - B_{post})} \times 100$$

Where the $D$ = donor intensity and $B$ = background intensity.

For BiFC experiments, expression and localization of VN and VC fusion proteins were determined by confocal microscopy following staining with polyclonal rabbit anti-GFP (1:1000 dilution; Affinity BioReagents, Golden, CO) and goat anti-GFP (1:500; Santa Cruz Biotechnology, Inc., Santa Cruz, CA) antibodies followed by anti-rabbit secondary Ab conjugated with Alexa Fluor 594 and anti goat secondary Ab conjugated with Alexa Fluor 405. BiFC microscopy was performed using a Zeiss LSM 700 confocal microscope equipped with a 63× oil immersion objective and appropriate filter set for the BiFC signal (green), and the VN (red) or VC (blue) fusion proteins.

## Lectin resistance test

Resistance to the lectins L-PHA and Con A was determined as described (*Stanley and Sundaram, 2014*). Briefly, 2000 CHO cells were added to each well of a 96-well plate in 100 µl culture medium, followed by 100 µl lectin at increasing concentrations and incubation at 37°C in a 5% $CO_2$ incubator. When control wells were confluent (~4 days), medium was removed and cells remaining attached to the plate were stained with Methylene Blue in 50% methanol.

## Flow cytometry and FACS

For flow cytometry, $5 \times 10^5$ cells were washed with 1 ml FACS binding buffer (Hank's buffered salt solution containing 1 mM $CaCl_2$, 1 mM $MgCl_2$, 0.05% or 0.1% sodium azide, and 2% BSA Fraction V [Sigma]) at 4°C and incubated with the mannose binding lectin from *G. nivalis* (GNA) conjugated to FITC at 12 µg/ml in FACS buffer on ice. After 30 min cells were washed with 1 ml FACS buffer, resuspended in 0.5 ml FACS buffer, without BSA, and **7**-Amino-actinomycin D (BD Biosciences) was added prior to analysis in a FACSscan (BD Biosciences) flow cytometer. Flowjo software (Tree Star Inc., Ashland, OR) was used to obtain profiles after 7-AAD-positive cells were gated out. For cell sorting, FACS binding buffer without sodium azide was used. Cells were resuspended in 0.5 ml FACS buffer containing penicillin (100 units) and streptomycin (100 µg/ml, Invitrogen) and amphotericin B (2.50 µg/ml, Invitrogen) and subjected to flow cytometry (DakoCytomation MoFlo and Dako MoFlo XDP, Beckman Coulter, Jersey City, NJ) to sort GFP- or GNA- binding cells and remove **7**-AAD-positive cells.

## Glycosyltransferase assays

Exponentially growing cells were washed three times and lysed ($10^7$ cells/75 µl) in 1.5% Triton X-100 in distilled water containing protease inhibitor cocktail (Roche, Nutley, NJ). MGAT1 and B4GALT1 were assayed as described previously (*Huang and Stanley, 2010*) using $Man_5GlcNA\text{-}c_2Asn$ and UDP-$^3$H-GlcNAc for MGAT1, and GlcNAc with UDP-$^3$H-Gal for B4GALT1. To determine MGAT2 and MGAT5 activities, synthetic glycan acceptors specific for MGAT2 (GlcNAcβ1,2Manα1,3(Manα1,6)Manβ1,4GlcNAcβ1,octyl) or (MGAT5 Manα1,6Manβ1,4GlcNAcβ1, octyl) respectively, were kindly provided by Dr Ole Hindsgaul. Assays were performed in a final volume of 50 µl containing ~50 µg cell lysate protein incubated in duplicate at 37°C for 2 hr with 20–40 µg substrate in the 62.5 mM 2-(*N* morpholino)ethanesulfonate (MES) (pH 6.25–6.5), 25 mM $MnCl_2$, and 0.75 mM UDP-[$^3$H]-GlcNAc (10,000–20,000 cpm/nmol; Perkin Elmer, Inc., Waltham, MA). MGAT1 reactions were stopped by adding 0.5 ml of Con A buffer (0.1 M sodium acetate, 1.0 M NaCl, 10 mM $MgCl_2$, 10 mM $CaCl_2$, 10 mM $MnCl_2$, and 0.02% sodium azide). After centrifugation in a microfuge, the supernatant was added to a 1 ml column of Con A-Sepharose (GE Healthcare, Piscataway, NJ). For MGAT1, the column was washed with Con A buffer and the product eluted with 200 mM α-methylmannoside in Con A buffer. For MGAT2 and MGAT5 assays, reaction products were

separated on a SepPak column to which the octyl moiety of the acceptor bound. Specific activities (nmol transferred per mg protein per hour) were determined from $^3$H-GlcNAc incorporated into products in the presence vs the absence of acceptor, or by comparison with boiled extract. ß4GalT activity was assayed using GlcNAc as acceptor as described (*Lee et al., 2001*).

## Western analysis

Transfectants were washed with PBS and lysed in distilled water containing 75 µl of 1.5% Triton X-100 (Sigma–Aldrich) with protease inhibitor cocktail (Roche) per $10^7$ cells. Protein was determined by Dc protein assay (Bio-Rad, Hercules, CA) and ~50–100 µg protein electrophoresed in a 10% Tris-HCl polyacrylamide gel at 10–30 mA for 2 hr. Transfer to polyvinylidene difluoride (PerkinElmer, Inc.) membrane was performed overnight at 50 mA in buffer containing 10% methanol. Antibodies were diluted in Tris buffered saline (10 mM Tris HCl, pH 7.4, 150 mM NaCl) containing 0.05% Tween 20 (Sigma Aldrich) and 3% nonfat dry milk supplemented with 3% BSA (Fraction V) or 3% nonfat dry milk, respectively. Antibody dilutions were: anti-Myc mAb (9E10) 1:500, anti-HA mAb (HA.11) 1:1000, anti-beta actin mAb 1:5000, and HRP-conjugated goat anti-mouse secondary antibody, 1:5000–10,000. After washing in Tris buffered saline containing 0.05% Tween 20, membrane was incubated with Super Signal West Pico chemiluminescence reagent (Thermo Scientific) and exposed to film (Denville Scientific, Inc., South Palinfield, NJ).

## Analysis of mouse RNA sequencing data

Mouse RNA sequencing data (*Soumillon et al., 2013a*) containing the FPKM values for all five germ cell subtypes were downloaded from the GEO database at http://www.ncbi.nlm.nih.gov/geo/query/acc.cgi?acc=GSE43717. The frequency of $\log_2$-transformed FPKM values of all 36,823 transcripts were displayed as histograms and the $\log_2$-transformed FPKM values for *Mgat1* (ENSMUSG00000020346) and GnT1IP/*Mgat4d* (ENSMUSG00000035057) mapped as vertical bars in order to visualize their gene expression levels. All analyses and visualizations were conducted using the statistical programming environment *R* (www.r-project.org; *Figure 9—source data 1*).

## Analysis of human microarray data

Human testis microarray data containing the $\log_2$-transformed fluorescence values from *Spiess et al. (2007)* were downloaded from the ArrayExpress database at http://www.ebi.ac.uk/arrayexpress/experiments/E-TABM-234/. Expression values for GnT1IP (*MGAT4D*; LOC152586, Probeset 1569995_at) and *MGAT1* (Probeset 201126_s_at) were extracted for all replicates of spermatogenic subtypes, $\log_2$-transformed, and displayed as boxplots. In detail, these were TA, SCO, presence of SPG, MA at the level of primary spermatocytes, hypospermatogenesis with decreased numbers of round/elongated spermatids (HYS), and full spermatogenesis with normal numbers of round/elongated spermatids (FS).

## Acknowledgements

We thank Dr Frank Batista for preparing the GnT1IP-L/MGAT1 construct and for helpful conversations. This work was supported by grants to PS from the National Institutes of Health (RO1 GM 105399 and RO1 NCI 36434), grants to SK from Oulu University and AH from the Emil Aaltonen Foundation, and Grant Sp721/4-1 from the German Research Foundation (DFG) to ANS. Partial support was provided by the Albert Einstein Cancer Center grant PO1 13330.

## Additional information

### Funding

| Funder | Grant reference | Author |
| --- | --- | --- |
| National Institutes of Health (NIH) | RO1 GM105399, RO1 CA36434, PO1 CA13330 | Pamela Stanley |
| Emil Aaltosen Säätiö (Emil Aaltonen Foundation) | | Antti Hassinen |

| Funder | Grant reference | Author |
|--------|----------------|--------|
| Deutsche Forschungsgemeinschaft (DFG) | Sp72¼-1 | Andrej-Nikolai Spiess |

The funders had no role in study design, data collection and interpretation, or the decision to submit the work for publication.

### Author contributions

H-HH, AH, Conception and design, Acquisition of data, Analysis and interpretation of data, Drafting or revising the article; SS, Acquisition of data, Analysis and interpretation of data; A-NS, Analysis and interpretation of data, Drafting or revising the article; SK, PS, Conception and design, Analysis and interpretation of data, Drafting or revising the article

# Additional files

### Supplementary file

• Source code 1. Glycosyltransferase assay data for CHO cells expressing HA-GnT1IP-L.

### Major datasets

The following previously published datasets were used:

| Author(s) | Year | Dataset title | Dataset ID and/or URL | Database, license, and accessibility information |
|-----------|------|---------------|------------------------|---------------------------------------------------|
| Soumillon M, Necsulea A, Weier M, Brawand D, Zhang X, Gu H, Barthès P, Kokkinaki M, Nef S, Gnirke A, Dym M, de Massy B, Mikkelsen TS, Kaessmann H | 2013 | Cellular source and mechanisms of high transcriptome complexity in the mammalian testis (RNA-Seq cells) | http://www.ncbi.nlm.nih.gov/geo/query/acc.cgi?acc=GSE43717 | Publicly available at the NCBI Gene Expression Omnibus (Accession no: GSE43717). |
| Cappallo-Obermann H, Feig C, Schulze W, Spiess AN, Reprod H, Spiess AN, Feig C, Schulze W, Chalmel F, Cappallo-Obermann H, Primig M, Kirchhoff C | 2008, updated 2014 | E-TABM-234 - Transcription profiling of human testis samples from men with highly defined and homogenous testicular pathologies reveals patterns that correlate with distinct stages of spermatogenesis | http://www.ebi.ac.uk/arrayexpress/experiments/E-TABM-234/ | Publicly available at the EBI European Nucleotide Archive (Accession no: E-TABM-234). |
| Fagerberg L, Hallström BM, Oksvold P, Kampf C, Djureinovic D, Odeberg J, Habuka M, Tahmasebpoor S, Danielsson A, Edlund K, Asplund A, Sjöstedt E, Lundberg E, Szigyarto CAK, Skogs M, Takanen JO, Berling H, Tegel H, Mulder J, Nilsson P, Schwenk JM, Lindskog C, Danielsson F, Mardinoglu A, Sivertsson Å, Felitzen K, Forsberg M, Zwahlen M, Olsson I, Navani S, Huss M, Nielsen J, Ponten F, Uhlén M | 2013, updated 2014 | E-MTAB-1733 - RNA-seq of coding RNA from tissue samples of 95 human individuals representing 27 different tissues in order to determine tissue-specificity of all protein-coding genes | https://www.ebi.ac.uk/arrayexpress/experiments/E-MTAB-1733/ | Publicly available at the EBI European Nucleotide Archive (Accession no: E-MTAB-1733). |

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
