## [Decision Letter]

Thank you for submitting your work entitled “GnT1IP-L specifically inhibits MGAT1
in the Golgi via its luminal Domain” for peer review at *eLife*.
Your submission has been favorably evaluated by Randy Schekman (Senior Editor), and
three reviewers, one of whom, Reid Gilmore, is a member of our Board of Reviewing
Editors, and another is John Hanover.

The reviewers have discussed their reviews with one another, and the Reviewing Editor
has drafted this decision to help you prepare a revised submission.

The manuscript from Huang et al. examines the mechanism of MGAT1 inhibition by the
GnT1P-L, a protein that appears to be a member of the MGAT gene family based upon
homology. Previous research form the Stanley lab demonstrated that overexpression of
GnT1P-L does not increase transfer of GlcNAc to N-glycans, but instead reduces MGAT1
activity. The current manuscript follows up on these earlier findings by testing whether
GnT1P-L can inhibit other MGATs. The authors use FRET and BiFC experiments to
characterize the homomeric and heteromeric interactions of GnT1P-L and MGAT1.
Importantly, GnT1P-L does not inhibit MGAT2-MGAT5, and does not interact with these
other GlcNAc-transferases. Heteromeric interactions between GnT1P-L and MGAT1 occur in
the Golgi and are dependent upon the slightly acidic environment of the Golgi lumen.
Analysis of MGAT1 and GnT11P-L expression in spermatocyte progenitor cells point to an
important role for GnT1P-L during spermatogenesis.

Essential revisions:

1) Figure 2: the first experiment in the
manuscript tests the effect of replacing phenylalanine residues in the TM span with
leucine or alanine residues. The rationale for mutating these residues is not that clear
since F, A and L residues are all typical amino acids for TM spans. The lack of a wild
type GnT1P-L control (Myc or HA tagged) in panel C is another weakness of this figure.
The reader can't tell whether the (F/L) or (F/A) mutants are as effective as the
parental construct. Oddly enough, the HA-tagged constructs are less effective than the
Myc-tagged constructs. The authors don't comment on this tag-induced activity
difference, as the next experiment (Figure 3)
shows that both the cytosolic domain and the TM span of GnT1P-L can be replaced with the
corresponding segments of MGAT1. Deletion of this figure would improve the manuscript
since the results don't make a significant contribution after Figure 3 is shown. Figure 1 should be modified to eliminate the diagrams for the F/A and F/L
constructs.

2) Figure 3: the authors should explain the
lectin growth inhibition assay better. Figure 3
would be improved by inclusion of wild type GnT1P-L and Lec1 cells. The main text needs
to indicate that cell colonies are detected by staining with methylene blue. The text
should also indicate that GnT1P-L expressing cells are not expected to show the same
level of resistance as Lec1 cells, since MGAT1 activity is reduced, not eliminated.

3) The authors should comment on the roughly 2-fold difference in observed FRET
efficiency for the same donor/acceptor pair in Figure 5 versus Figure 5. Is this explained
by a difference in MGAT1 expression? The authors should consider changing the lettering
for panel D to stress that the constructs tested (e.g., +MGAT1) are HA tagged,
and are being tested as competitive inhibitors.

4) Previous publications from Hassinen et al. have described Golgi-localized MGAT1-MGAT2
heteromers. The authors should test whether GnT1IP-L expression reduces MGAT1-MGAT2
heteromer formation using FRET and/or BiFC. This would provide insight into whether
GnT1IP-L dissociates MGAT1 from MGAT2-MGAT1 heteromers, or instead exerts inhibitory
activity within the context of pre-existing MGAT1-MGAT2 complexes. I believe the authors
have the necessary cell lines and expression constructs to conduct this experiment.

5) The final section of the manuscript makes an abrupt change to transcript profiling of
GnT1IP-L and MGAT1 transcripts in mouse germ cells and male testis biopsies. This latter
section of the manuscript was based on mining of previous microarray and RNA-Seq data
sets for GnT1IP-L and MGAT1 transcript levels, but the data presentation and description
of methods employed are cryptic or entirely missing. No descriptions of the informatics
approaches for transcript profiling are presented in the Methods section and only
literature references to the original data sets are listed in the figure legends. In
addition, the descriptions in the figure legends and labeling of the figures (Figure 10 and 11) were incomplete or misleading.
It is not clear from the figure or legend that Figure 10 is RNA-Seq data and the plot is labeled as “log2
fluorescence,” when it is likely supposed to be labeled “log 2
FPKM.” FPKM needs to be defined. A much more explicit description of the methods
is required for Figure 10 and the axes need to
be labeled in a comprehensible manner. The data in Figure 11 is also cryptically
presented and it is not clear until the last line of the legend that the authors are
describing a RNA microarray experiment. The connection between the transcript profiling
and the remainder of the manuscript seems rather tenuous, but an effort is made to link
protein expression with glycan phenotypes with glycan structure phenotypes in the
respective cell types and human pathology of defects in spermatogenesis. This link from
transcript levels to the glycan structures in the respective cell types needs to be more
clearly presented so that it is clear what the take home message is for the gene
expression data. Observations are made that GnT1IP-L expression is elevated in
postmeiotic germ cells consistent with data in Figure 11 on human testis biopsies from
men with impaired spermatogenesis. The authors hypothesize that blockage in glycan
maturation is critical for germ cell interaction with Sertoli cells.

6) In the Discussion, the authors indicate that the previous paper (Huang et al., 2010)
showed that the stem deletion mutant (delta-stem-GNT1IP-L) and the C-terminal deletion
mutant of GNT1IP-L-CD1 still interact with MGAT1 but do not cause inhibition. For
GNT1IP-L-CD1, Figure 4 of Huang et al. showed
that the C-terminal deletion mutant is mainly retained in the ER, so formation of
GNT1IP-L-CD1-MGAT1 heteromers should be reduced based upon the results presented in
Figure 7 of the current manuscript. Unless I
have overlooked something, interaction between a C-terminal GNT1IP deletion mutant and
MGAT1 has only been tested in the context of the short form (GNT1IP-S-CD2 in Figure 5). Since the short form (GNT1IP-S) is not a
MGAT1 inhibitor, one has to be concerned that ER retention of the truncation mutant is a
contributing factor in lack of inhibitory activity. Since the current manuscript does
not use BiFC or FRET to characterize formation of heteromers between MGAT1 and these
inactive GNT1IP-L mutants, it seems premature to conclude that specific sub-regions of
the lumenal domain are required for GNT1IP-L inhibitory activity.

---

## [Author Response]

*1)*
Figure 2*: the first
experiment in the manuscript tests the effect of replacing phenylalanine residues in
the TM span with leucine or alanine residues. The rationale for mutating these
residues is not that clear since F, A and L residues are all typical amino acids for
TM spans*.

Our rationale was based on the relative hydrophobicity index of F, A and L. F and L have
a similarly high hydrophobicity index whereas Ala has an ∼50% lower
hydrophobicity index. We reasoned that if we changed 5 Phe residues to 5 Ala residues
this would be a significant and potentially functional change, whereas changing to 5 Leu
residues should have minimal effect, acting as a positive control. We have included this
rationale in the text and added the lectin resistance data for Myc-GnT1IP/L(F/L) and
Myc-GnT1IP-L(F/A) to revised Figure 2.

*The lack of a wild type GnT1P-L control (Myc or HA tagged) in panel C is another
weakness of this figure. The reader can't tell whether the (F/L) or (F/A)
mutants are as effective as the parental construct. Oddly enough, the HA-tagged
constructs are less effective than the Myc-tagged constructs. The authors
don't comment on this tag-induced activity difference, as the next experiment
(*Figure 3*)
shows that both the cytosolic domain and the TM span of GnT1P-L can be replaced with
the corresponding segments of MGAT1. Deletion of this figure would improve the
manuscript since the results don't make a significant contribution
after*
Figure 3
*is shown.*
Figure 1
*should be modified to eliminate the diagrams for the F/A and F/L
constructs*.

The L-PHA resistance test of a hygromycin-resistant transfectant population is not a
directly quantitative test as it depends on the level of expression of a transgene in
relation to hygromycin resistance. The most important parameter reflecting GnT1IP-L
activity is whether a consistent proportion of transfectants survive high concentrations
of L-PHA. Transfectant populations sorted for high GnT1IP-L expression or oligomannose
expression show uniform resistance to L-PHA, as shown in our previous paper. We have now
included an explanation in the text. Our previous paper also rigorously tested different
tags on GnT1IP-L and found no consistent difference in HA versus Myc with respect to
GnT1IP-L inhibitory activity. We have removed Figure 2 and altered Figure 1 as requested.
However, we mention the results of the mutation experiment as the prelude to testing the
GnT1IP-L luminal domain for inhibitory activity, and include some data in revised Figure 2.

*2)*
Figure 3*: the authors
should explain the lectin growth inhibition assay better.*
Figure 3
*would be improved by inclusion of wild type GnT1P-L and Lec1 cells. The main
text needs to indicate that cell colonies are detected by staining with methylene
blue. The text should also indicate that GnT1P-L expressing cells are not expected to
show the same level of resistance as Lec1 cells, since MGAT1 activity is reduced, not
eliminated*.

We have expanded our description of the L-PHA resistance test and its interpretation as
requested. We also included Lec1 in revised Figure 2. Since we did not test a wild-type control in every plate, we present a
Myc-GnT1IP-L(F/L) control that was included in an experiment in which MGAT1/GnT1IP-L-Myc
was also tested.

*3) The authors should comment on the roughly 2-fold difference in observed FRET
efficiency for the same donor/acceptor pair in*
Figure 5
*versus*
Figure 5*. Is this
explained by a difference in MGAT1 expression? The authors should consider changing
the lettering for panel D to stress that the constructs tested (e.g., +MGAT1)
are HA tagged, and are being tested as competitive inhibitors*.

The lower FRET efficiency of mouse constructs likely reflects species differences as we
observed lower expression levels of all mouse MGAT enzyme constructs compared to their
human counterparts. The reason for this is unclear. However, if FRET efficiencies are
expressed as a percentage of MGAT1/GnT1IP-L interaction, no differences are evident
between mouse and human transferases. We have included a statement to this effect in the
revised text. The lettering has been changed as requested in revised Figure 4.

*4) Previous publications from Hassinen et al. have described Golgi-localized
MGAT1-MGAT2 heteromers. The authors should test whether GnT1IP-L expression reduces
MGAT1-MGAT2 heteromer formation using FRET and/or BiFC. This would provide insight
into whether GnT1IP-L dissociates MGAT1 from MGAT2-MGAT1 heteromers, or instead
exerts inhibitory activity within the context of pre-existing MGAT1-MGAT2 complexes.
I believe the authors have the necessary cell lines and expression constructs to
conduct this experiment*.

The suggested experiments have been added to revised Figure 4 (panels E and F) and the figure legend modified accordingly. No
differences were observed in MGAT1/MGAT2 interaction upon co-expression of a competing
GnT1IP-L-HA construct with both human and mouse transferase constructs. This is
consistent also with the inability of MGAT2-HA to inhibit MGAT1/GnT1IP-L interaction
(see Figure 4).

*5) The final section of the manuscript makes an abrupt change to transcript
profiling of GnT1IP-L and MGAT1 transcripts in mouse germ cells and male testis
biopsies. This latter section of the manuscript was based on mining of previous
microarray and RNA-Seq data sets for GnT1IP-L and MGAT1 transcript levels, but the
data presentation and description of methods employed are cryptic or entirely
missing. No descriptions of the informatics approaches for transcript profiling are
presented in the Methods section and only literature references to the original data
sets are listed in the figure legends. In addition, the descriptions in the figure
legends and labeling of the figures (Figures 10 and 11) were incomplete or
misleading. It is not clear from the figure or legend that*
Figure 10
*is RNA-Seq data and the plot is labeled as “log2 fluorescence,”
when it is likely supposed to be labeled “log 2 FPKM.” FPKM needs to be
defined. A much more explicit description of the methods is required for*
Figure 10
*and the axes need to be labeled in a comprehensible manner. The data in Figure
11 is also cryptically presented and it is not clear until the last line of the
legend that the authors are describing a RNA microarray experiment. The connection
between the transcript profiling and the remainder of the manuscript seems rather
tenuous, but an effort is made to link protein expression with glycan phenotypes with
glycan structure phenotypes in the respective cell types and human pathology of
defects in spermatogenesis. This link from transcript levels to the glycan structures
in the respective cell types needs to be more clearly presented so that it is clear
what the take home message is for the gene expression data. Observations are made
that GnT1IP-L expression is elevated in postmeiotic germ cells consistent with data
in Figure 11 on human testis biopsies from men with impaired spermatogenesis. The
authors hypothesize that blockage in glycan maturation is critical for germ cell
interaction with Sertoli cells*.

We have modified the text to introduce the context for presenting the gene profiling
data and to include a take home message. We have also revised Figures 10 and 11 (now
Figures 9 and 10) and their legends,
added information to the Materials and methods section and rephrased the Results.
Materials and methods now includes two sections: “Analysis of mouse RNA
sequencing data” and “Analysis of human microarray data”, in which
we describe in detail how datasets were downloaded, which Reads/Probesets were
extracted, and how they were visualized with respect to germ cell subtypes and
testicular phenotypes. The legends have been updated so that the legend title now
contains data origin (RNA-Seq or Microarray). The x-axis of the histogram now has the
correct label “log2 FPKM” (thanks for alerting us). Also FPKM is now
defined in the legend to the new Figure 9 as well
as in the Materials and methods section.

*6) In the Discussion, the authors indicate that the previous paper (Huang et
al., 2010) showed that the stem deletion mutant (delta-stem-GNT1IP-L) and the
C-terminal deletion mutant of GNT1IP-L-CD1 still interact with MGAT1 but do not cause
inhibition. For GNT1IP-L-CD1,*
Figure 4
*of Huang et al. showed that the C-terminal deletion mutant is mainly retained in
the ER, so formation of GNT1IP-L-CD1-MGAT1 heteromers should be reduced based upon
the results presented in*
Figure 7
*of the current manuscript. Unless I have overlooked something, interaction
between a C-terminal GNT1IP deletion mutant and MGAT1 has only been tested in the
context of the short form (GNT1IP-S-CD2 in*
Figure 5*). Since the
short form (GNT1IP-S) is not a MGAT1 inhibitor, one has to be concerned that ER
retention of the truncation mutant is a contributing factor in lack of inhibitory
activity. Since the current manuscript does not use BiFC or FRET to characterize
formation of heteromers between MGAT1 and these inactive GNT1IP-L mutants, it seems
premature to conclude that specific sub-regions of the lumenal domain are required
for GNT1IP-L inhibitory activity*.

We have added a description of the similarities and differences between GnT1IP-S and
–L to the Introduction and elsewhere. Our previous paper showed that adding an
N-terminal tag (Myc or HA) to GnT1IP-S converts it to a membrane-bound form that
inhibits MGAT1 in transfected cells, indistinguishable from GnT1IP-L. The sequence of
GnT1IP-L from aa 45 to 417 is identical to GnT1IP-S and differs only in that it has a 44
aa N-terminal extension. Some mutant constructs were made with Tag-GnT1IP-S which
inhibits MGAT1 (Table 1, [16]), and mislocalizes co-expressed MGAT1 to
the ER more severely than Myc-GnT1IP-S-CD1 (C-terminal 39 aa deletion) (Figure 4 in [16]). Tag-GnT1IP-S-CD1 does not induce resistance to L-PHA
(Table 1), nor inhibit MGAT1 activity. Thus,
it is appropriate to say that loss of the 39 C-terminal amino acids inactivates
inhibition of MGAT1 by membrane-bound GnT1IP. Co-immunoprecipitation with MGAT1 was
observed with a 121 aa C-terminal deletion, which suggests that the 39 aa deletion
mutant would also interact with MGAT1. We have modified the revised text.